



# Airborne investigation of black carbon interaction with low-level, persistent, mixed-phase clouds in the Arctic summer

Marco Zanatta[1,2], Stephan Mertes[3], Olivier Jourdan[4], Regis Dupuy[4], Emma Järvinen[2], Martin Schnaiter[2], Oliver Eppers[5,6],
Johannes Schneider[5], Zsófia Jurányi[1], Andreas Herber[1]

[1]Alfred-Wegener-Institut, Helmholtz-Zentrum für Polar- und Meeresforschung (AWI), Bremerhaven, Germany

[2] Institute of Meteorology and Climate Research, Karlsruhe Institute of Technology, Karlsruhe, Germany

[3]Leibniz-Institut für Troposphärenforschung, Leipzig, Germany

[4] Laboratoire de Météorologie Physique, Université Clermont Auvergne/OPGC/CNRS, UMR 6016, Clermont-Ferrand, France

[5]Particle Chemistry Department, Max Planck Institute for Chemistry, Mainz, Germany

[6] Institute for Atmospheric Physics, University of Mainz, Mainz, Germany

*Correspondence to*: Marco Zanatta (marco.zanatta@kit.edu)

**Abstract.** Aerosol-cloud interaction is considered one of the largest sources of uncertainties in radiative forcing estimations. To better understand the role of black carbon aerosol as cloud nucleus and the impact of clouds on its vertical distribution in the Arctic, we report airborne in-situ measurements of refractory black carbon aerosol particles (rBC) in the European Arctic near Svalbard during the ACLOUD campaign held in summer 2017. rBC was measured with a single particle soot photometer on board of the research aircraft "Polar 6" from the lowest atmospheric layer up to approximately 3500 m asl. During in-cloud flight transects, rBC particles contained in liquid droplets (rBC residuals) were sampled through a counterflow virtual impactor (CVI). Overall, the presence of low-level clouds was associated with a radical change in the concentration and size distribution of rBC particles in the boundary layer compared to the free troposphere. Four flights conducted in the presence of inside-inversion, surface-coupled, mixed-phase clouds over sea ice, were selected to address the variability of rBC particles sampled above, below and within the cloud layer. We show that the properties of rBC such as concentration, size and mixing state drastically changed from the above to the below cloud layers, but also within the cloud layers from cloud top to cloud bottom. Our results might suggest the occurrence of a cloud-mediated transformation cycle of rBC particles in the boundary layer which includes activation, cloud processing, and sub-cloud release of processed rBC agglomerates. In the case of persistent low-level Arctic clouds, this cycle may reiterate multiple times, adding one additional degree of complexity to the understanding of cloud processing of black carbon particles in the Arctic.



# 1    Introduction

It is well known that clouds strongly impact the surface energy budget in the Arctic (Shupe and Intrieri, 2004; Lubin and Vogelmann, 2006). However, the interaction between aerosol particles and clouds remains one of the major sources of uncertainty in radiative forcing estimations, not only in the Arctic but on a global scale as well (Bellouin et al., 2020; IPCC et al., 2021). In fact, aerosol particles may influence the radiative properties of Arctic clouds by controlling the microphysical properties of liquid droplets (Earle et al., 2011; McFarquhar et al., 2011; Coopman et al., 2018) and ice formation (Solomon et al., 2018; Eirund et al., 2019). Otherwise, cloud formation and subsequent precipitation were found to be key mechanisms controlling the aerosol seasonal cycle in the Arctic atmosphere (Croft et al., 2016; Willis et al., 2018).

Black carbon (BC) particles are carbonaceous aerosol particles emitted by incomplete combustion of fossil fuels and biomass (Bond et al., 2004) and, due to their unique absorption of solar radiation in the visible spectrum, they play an important role in the Arctic radiative balance. First, BC particles directly interact with the solar radiation causing a net warming of its atmospheric layer (Flanner, 2013). Second, the change in local temperature might influence cloud's vertical distribution via semi-direct effects (Sand et al., 2013). Third, via aerosol-cloud interaction, also known as the first indirect effect, black carbon activation might lead to a change in the microphysical properties of clouds with consequences on cloud radiative properties (Sand et al., 2013). Lastly, after deposition, BC can decrease the snow albedo promoting melting via the snow darkening effect (Flanner et al., 2009). All these processes are strongly interconnected (Quinn et al., 2015), making BC the second atmospheric Arctic warmer only after the trace gases carbon dioxide and methane (Oshima et al., 2020).

The ability of BC to interact with clouds is particularly relevant, since cloud scavenging is one of the factors controlling the temporal and spatial variability of BC in the Arctic atmosphere and snow. Precipitation occurring during long-range transport influences the seasonal cycle of BC, with the highest precipitation rate in summer leading to a decline of BC burden from late spring to autumn (Garrett et al., 2011; Mori et al., 2020). Moreover, wet deposition influences the vertical distribution of BC in the atmosphere, with convective precipitation controlling the concentration of BC in the upper troposphere and stratiform precipitation controlling the concentration of BC at the surface (Mahmood et al., 2016). Considering that wet scavenging is responsible of 90% of BC mass deposition in the Arctic (Dou and Xiao, 2016), aerosol-cloud interaction, might easily be considered the process limiting the aerosol-radiation forcing of BC. In fact, removal of surface BC by stratiform clouds (Mahmood et al., 2016) might reduce aerosol-radiation forcing at the surface (Flanner, 2013), but increase snow darkening (Flanner et al., 2009). The complexity of in-cloud and below-cloud scavenging (Hegg et al., 2011; Gogoi et al., 2018) and snow post-processing (Doherty et al., 2013) makes, however, these estimations extremely uncertain (Kang et al., 2020), especially because global models have limited ability to reproduce temporal, vertical and horizontal distribution of BC in the whole Arctic region (Whaley et al., 2022). The first limitation is represented by the parametrization of BC particles as cloud seed.

Currently, there are no direct observations of BC activation of clouds in the Arctic region, and the aforementioned studies are mostly based on global model simulations and parametrizations derived from observations outside the Arctic region, indicating





that the ability to form liquid droplets in warm conditions depends on the size distribution, and on the degree of internal mixing

of BC particles. Fresh and externally mixed BC particles are not hygroscopic (Bond et al., 2013), ambient (Motos et al., 2019b) and laboratory experiments (Dalirian et al., 2018) showed that larger and thickly coated BC particles act as efficient cloud condensation nuclei. Moreover, the size dependent activation ratio of BC strongly depends on the ambient supersaturation (Motos et al., 2019a). At temperatures below the freezing point, the ability of BC to act as an ice nucleating particle is more uncertain. In mixed-phase clouds at ambient temperatures between -26°C and -8°C, the ice active fraction of BC particles

remained below 0.1% (Kupiszewski et al., 2016). However, the same study showed that ice activation fraction of BC increases as function of diameter and degree of internal mixing. Laboratory studies further confirmed that BC particles do not activate ice crystals at temperatures warmer than -38°C (Kanji et al., 2020). The understanding of BC activation in liquid and mixed-phase clouds becomes additionally complicated by cloud processing such as droplet coalescence, riming, Wegener–Bergeron–Findeisen (WBF) process and capture of interstitial aerosol, which modify the properties of BC particles and its partitioning

between the liquid, the ice and the interstitial phases (Verheggen et al., 2007; Pierce et al., 2015; Qi et al., 2017; Ding et al., 2019).

Despite the climatic impact of BC in the Arctic, in-situ observations are still very rare, especially at cloud level (Tørseth et al., 2019). In this work, we will present unprecedented airborne vertically resolved measurements of black carbon particles sampled in and outside clouds in the European Arctic in summer 2017 during the Arctic CLoud Observations Using airborne

measurements during polar Day (ACLOUD) campaign (Wendisch et al., 2018, 2022). The objective is to provide first insights on the BC-cloud interaction in the Arctic investigating the potential correlation between cloud presence and its phase with the vertical distribution of BC properties.

## 2    Methodology

The ACLOUD campaign was conducted between 23 May and 26 June 2017 in the north west region of Svalbard (Norway)

within the framework of the "Arctic Amplification: Climate Relevant Atmospheric and Surface Processes, and Feedback Mechanisms (AC)3" project (Wendisch et al., 2018, 2022), see https://www.ac3-tr.de/. All validated data are published in the World Data Center PANGAEA as instrument-separated data subsets (Ehrlich et al., 2019b, https://doi.org/10.1594/PANGAEA.902603). Flight operations and atmospheric measurements used in this specific work are described in the following, while instrumentation, measured parameters and relative abbreviations are listed in Table 1.

## 2.1    Flight operations

Atmospheric observations were carried out with two research aircrafts of the Alfred Wegener Institute (AWI), Polar 5 and Polar 6 (Wesche et al., 2016). Polar 5 was equipped with remote-sensing instruments, while Polar 6 was equipped with in-situ measurements. A full list of the deployed instrumentation can be found in Ehrlich et al. (2019). In this work, we will present the results obtained from three subsets of whole 22 research flights. The first subset included 12 flights performed on board of



Polar 6 providing the most complete vertical coverage of aerosol and cloud particles up to and altitude of 3500 m asl. The flights were performed on 27 May and on 02, 04, 05, 08, 13, 14, 16, 17, 18, 23, 26 June over open water, marginal sea ice zone, and ground (Figure 1a). The second subset was composed by 4 flights performed on 02, 04, 05, 08 June with repeated sampling of low level clouds over the marginal sea ice zone (Figure 1b), thus providing the best opportunity to investigate the interaction of black carbon with mixed-phase clouds in the boundary layer. The last subset was composed by one single flight

of Polar 6, performed on 25 June, which provided the sole case of clear-sky conditions (Figure 1b).

## 2.2    Techniques

### 2.2.1    Meteorological measurements

Meteorological parameters such as pressure, humidity and temperature were recorded at 1 Hz resolution with the basic meteorological sensor suite of Polar 6 fully described in previous works (Herber et al., 2012; Schulz et al., 2019; Ehrlich et al.,

2019). T and RH data were merged with aircraft position and air pressure into a 1 Hz basic meteorological dataset (https://doi.org/10.1594/PANGAEA.902849; Hartmann et al., 2019). The potential temperature ($T_P$) was calculated from measured ambient temperature ($T$) and ambient pressure ($P$) as: $T_P = T (P_0/P)^{0.286}$.

### 2.2.2    Cloud particle measurements

The Small Ice Detector Mark 3 (SID-3) records the spatial distribution of forward-scattered light from single cloud particles as 2-D scattering patterns (Hirst et al., 2001; Vochezer et al., 2016). The SID-3 allows deriving the cloud particle number size distribution and the resulting number concentration ($N_{Dro}$) and diameter ($D_{Dro}$) in the 5 - 45 µm diameter range (https://doi.org/10.1594/PANGAEA.900261; Schnaiter and Järvinen, 2019b). In the present work, this range was reduced to 10 - 45 µm. The liquid water content (LWC) was calculated from the size distribution assuming droplet sphericity and a water

density of 1 g cm$^{-3}$. LWC derived from SID-3 was then compared with the bulk LWC measured by a standard Nevzorov heated wire probe assuming a collection efficiency of 100% (Korolev et al., 1998). For the majority of clouds presented in this study, the LWC derived from the SID-3 data agreed with the LWC measured from the Nevzorov probe well within 10% (Figure S 1). Considering the good agreement with the LWC measured with the Nevzorov probe and the low concentration of ice crystals, all the cloud particles detected by the SID-3 were considered to be liquid droplets. Although not extensively

presented here, LWC data derived from the Nevzorov instrument are accessible at PANGAEA (https://doi.org/10.1594/PANGAEA.906658; Chechin, 2019).

The Cloud Imaging Probe (CIP, DMT, Longmont, CO, USA; Baumgardner et al., 2001) is based on the acquisition of two-dimensional black and white images of particles via the linear array technique which allows quantifying the dimension and shape of cloud particles. Due to the large uncertainties in the probe's sensitive area for the smallest particle sizes, the nominal

detection size range was reduced in the present work to 75 - 1550 µm. The number concentration of non-spherical ice crystals



($N_{Ice}$) was calculated according to circularity as in Crosier et al. (2011). The ice water content (IWC) was calculated with the mass-diameter relationship defined by Brown and Francis (1995). The mass fraction of ice water (IWF) was calculated as the ratio of IWC over the total water content (TWC). The latter was calculated as the sum of IWC measured by the CIP and LWC measured by the SID-3. It must be noted that the size detection range of the SID-3 and CIP did not overlap. Although the size-

segregated LWC derived from the SID-3 well represents the bulk LWC derived from the Nevzorov instrument, the contribution to IWC of ice crystals smaller than 75 µm and the contribution to LWC by droplets larger than 41 µm might lead to an unknown underestimation of IWF. Considering the number size distribution of liquid droplets and low number concentration of ice crystals, the underestimation of IWF was considered to be negligible. The number size distribution of ice crystals presented in this work is based on effective equivalent diameter, which is more comparable with the other microphysical probe installed on

Polar 6 during ACLOUD (CDP-2) and previous Arctic measurements (Mioche et al., 2017). Hereafter, equivalent diameter of ice crystals detected by CIP will be abbreviated as $D_{Ice}$. All ice crystals variables derived from CIP measurements are published in the PANGAEA database (https://doi.org/10.1594/PANGAEA.899074; Dupuy et al., 2019).

The Particle Habit Imaging and Polar Scattering (PHIPS) probe was used to identify habits of ice crystals and other morphological features. The PHIPS is a combination of a polar nephelometer and a stereomicroscopic imager recording

angular scattering functions and acquiring a bright field stereo-microscopic image of single particles in the size range 20-700 µm. More technical details on the basic principles, operation and validation of the PHIPS probe can be found in Schön et al. (2011), Abdelmonem et al. (2016), Schnaiter et al. (2018) and Waitz et al. (2021). Cloud particle habit classification was performed manually. In this study, we show statistical analysis of the fraction of rimed and aggregated ice crystals. The PHIPS data set is available in PANGAEA (https://doi.pangaea.de/10.1594/PANGAEA.902611; Schnaiter and Järvinen, 2019a).

The Airborne Mobile Aerosol Lidar (AMALi) system installed on board of Polar 5 was used to derive the cloud top height. Previous works provide technical details on the operation principle (Stachlewska, 2005), data processing for noise reduction (Stachlewska et al., 2010) and Arctic airborne deployment (Nakoudi et al., 2020). Briefly, the AMALi is a backscatter multichannel lidar covering the UV (1 unpolarized channel) and visible region (1 perpendicular and 1 parallel polarized channel). Downward probing of the atmosphere allowed identifying clouds below the aircraft from the attenuated backscatter

coefficients in the 532 nm parallel channel. Cloud conditions were defined for backscattering coefficients 5 times higher than cloud-free sections. Cloud top height with a 10 seconds time resolution are available on PANGAEA database (https://doi.org/10.1594/PANGAEA.899962; Neuber et al., 2019). We will solely discuss the cloud top height data acquired during collocated flights of Polar 5 with Polar 6 occurred on 27-29 June and 02-05-08-13-17 June.

### 2.2.3   Aerosol particle measurements

All aerosol particle data presented in this work were acquired with online single-particle instruments. A Single-Particle Soot Photometer (SP2, version D with 8-channels) by Droplet Measurement Technologies (DMT, Longmont, CO, USA) was used to detect refractory black carbon particles (rBC), following the terminology defined by Petzold et al. (2013). While a brief description of the operating principles of the SP2 and assumptions used in this study is given in the following, comprehensive





description of calibration standards and procedure is given by Moteki and Kondo (2010), Gysel et al. (2011) and Laborde et
al. (2012). By laser-induced incandescence, the SP2 is capable of quantifying the mass of absorbing and refractory material
contained in aerosol particles passing through the high-intensity continuous-wave, intra-cavity laser beam at a wavelength of
1064 nm (Stephens et al., 2003). The incandescence light detector was calibrated with a fullerene soot standard from Alfa
Aesar (stock no. 40971, lot no. FS12S011), size selected with a differential mobility analyser (SMPS; TSI, Shoreview, MN,
USA). The SP2 installed on the Polar 6 provided the number concentration ($N_{rBC}$), mass concentration ($M_{rBC}$) and size
distribution of rBC particles in the 0.37 - 178 fg mass range, converted to a mass equivalent diameter ($D_{rBC}$) range of 70 - 575
nm using a fixed bulk (void-free) density of 1800 kg m$^{-3}$ (Moteki et al., 2010). The optical diameter of rBC-free particles was
inferred from the scattering signal acquired by avalanche photodetectors in the given solid angles under the assumption of
spherical particles and a refractive index of 1.50+0i with Mie theory (Bohren and Huffman, 1998). Calibration of the scattering
cross section measurements was done using monodisperse spherical polystyrene latex (Thermo Scientific). The leading-edge
only technique was applied to estimate the coating thickness of rBC containing particles from the optical diameter of
unperturbed rBC cores and rBC-containing particles (Gao et al., 2007). The refractive index of the coating was assumed to be
equal to BC-free particles (1.50 + 0i) while the refractive index of rBC cores was set to be 1.90+0.8i. This value, being lower
than previous Arctic studies (2.26 + 1.26i; Raatikainen et al., 2015; Kodros et al., 2018; Zanatta et al., 2018) and higher than
measurements in continental Europe (1.75 + 0.43i; Yuan et al., 2021), allowed the mass equivalent diameter of rBC core to
match its optical diameter. The coating thickness was quantified for rBC particles having a $D_{rBC}$ between 200 nm and 300 nm.
It must be noted that the scattering detector failed on 07 June 2017; hence no LEO-fit analysis was performed on the following
flights. rBC data acquired with the SP2 are publicly available at https://doi.pangaea.de/10.1594/PANGAEA.899937 with the
temporal resolution of 3s (Zanatta and Herber, 2019b).

The Ultra High Sensitivity Aerosol Spectrometer (UHSAS; DMT, Longmont, CO, USA) measured the number concentration
($N_{AP}$) and size distribution of aerosol particles in the optical diameters range of 60-1000 nm (Cai et al., 2008). The UHSAS
was connected in parallel to SP2 at tubing length distance of 15 cm. Due to low signal-to-noise ratio at small sizes, the
concentration and size distribution estimated from the UHSAS are valid within the optical diameter range of 80-1000 nm
(Zanatta et al., 2020). It must be noted that rapid change of pressure might affect the sample flow measurement and,
consequently, the quantification of aerosol particle number concentration by the UHSAS (Brock et al., 2011). Although
modification of the UHSAS flow system are recommended for airborne operation (Kupc et al., 2018), Schulz et al. (2019)
showed no measuring bias of unmodified UHSAS during low-speed flights installed in the unpressurized cabin of Polar 6.
Aerosol particles data acquired with the UHSAS are publicly available at https://doi.pangaea.de/10.1594/PANGAEA.900341
with the temporal resolution of 3s (Zanatta and Herber, 2019a).

## 2.3 Cloud and aerosol particles sampling

Two different inlets were installed on the front top of the aircraft Polar 6, ahead of the engines to sample the total aerosol and
cloud particle residuals. A comprehensive description of the two inlets is given by Ehrlich et al. (2019).





The total aerosol inlet was a stainless-steel inlet with a shrouded diffuser already installed on Polar 6 in previous Arctic campaigns (Leaitch et al., 2016; Schulz et al., 2019). The manifold exhaust flowed freely into the back of the cabin, such that the intake flow varied with the true airspeed of the aircraft. Sampling speed at the inlet tip was approximately isokinetic for the airspeeds during ACLOUD, leading to a near-unity transmission of submicrometric aerosol particles (Ehrlich et al., 2019). SP2 and UHSAS shared one bypass line off the main aerosol inlet. All the aerosol particles data representative of clear-sky conditions include SP2 and UHSAS measurements performed only at the aerosol inlet in the absence of cloud. Clear-sky conditions were defined by $N_{Dro} = 0$ cm$^{-3}$ and LWC = 0 g m$^{-3}$.

A counterflow virtual impactor (CVI; Ogren et al., 1985; Noone et al., 1988) allowed size selective sampling of cloud particles by use of a counterflow at the inlet tip. During the in-cloud observation periods presented here, a counterflow between 3-4 L min$^{-1}$ ensured the sampling of cloud particles larger than approximately 10 µm, while all smaller cloud and aerosol particles were decelerated, stopped, and blown out of the inlet by the counterflow. Cloud residual particles were then released following evaporation or sublimation of liquid droplets or ice crystals, respectively. Hence, cloud particle residuals were representative of cloud condensation nuclei and/or ice nucleating particles (Mertes et al., 2005, 2007). In order to calculate the concentration of cloud particle residuals, the enrichment factor (EF) needed to be considered. EF was calculated as the ratio between the air volume flows in front and within the CVI, which varied between a minimum of 3.2 and maximum of 5.4 with a median value of 4.2. The transmission efficiency (TE) within the CVI inlet was calculated, similar to Schroder et al. (2015), as the ratio of the number of droplets larger than CVI size cut-off (10 µm), measured by the SID-3 over the number concentration of aerosol particles measured by the UHSAS in the optical diameter range of 80-1000 nm and corrected by the enrichment factor of CVI. Overall, TE varied between a flight average minimum of 16% (05 June) to a maximum of 23% (on 08 June), with an overall median value of 21%. Finally, the number concentration of rBC in cloud particle residuals ($N_{rBC-res}$) was calculated as:

$$N_{rBC-res} = \frac{N_{rBC}}{EF \times TE} \qquad\qquad 1$$

All cloud residuals data presented in this work refers to valid SP2 and UHSAS measurements at the CVI inlet only during in-cloud conditions: $N_{Dro} > 10$ cm$^{-3}$ and LWC > 0.01 g m$^{-3}$.

## 3 Results

### 3.1 Overview of vertical distribution of rBC and cloud particles during ACLOUD

Twelve ACLOUD flights were selected to investigate the vertical profile of rBC and cloud particles marginal sea ice zone, open water and land in the north-west of Svalbard (Figure 1a), and covered the three synoptic conditions identified during the ACLOUD campaign (Knudsen et al., 2018). One flight (27 May) was performed during the "cold period" when cold and dry





conditions were dominant, four flights (02, 04, 05, 08 June) were performed during the "warm period" in presence of moist air, while seven flights (13, 14, 16, 17, 18, 23, 26 June) were affected by a mixture of air masses, "normal period".

The vertical variability of rBC mass concentration ($M_{rBC}$) and mass size distribution is shown in Figure 2a and Figure 2b, respectively. For this specific analysis, the in-cloud measurement periods were excluded. Over all altitudes, the median $M_{rBC}$
was 2.3 ng m$^{-3}$ with interquartile range of 0.86-4.8 ng m$^{-3}$. The median $M_{rBC}$ decreased from 5.2 ng m$^{-3}$ during the cold period to 3.2 ng m$^{-3}$ in the warm period and reached a minimum median value of 1.7 ng m$^{-3}$ during the normal period. This low concentration is expected during summer in the European Arctic (Roiger et al. 2015), in the Canadian Arctic (Schulz et al. 2019) and the Alaskan Arctic (Schwarz et al., 2013), and are connected with limited south-north circulation of airmasses (Bozem et al., 2019) and efficient wet removal south of the polar dome (Croft et al., 2016). Although the impact of pollution
plumes is not infrequent in the free-troposphere in the summer Arctic (Brock et al., 2011; Kupiszewski et al., 2013; Roiger et al., 2015), the average vertical profile of $M_{rBC}$ did not show any relevant pollution plume above 500 m asl, where $M_{rBC}$ median concentration varied between 1.7 ng m$^{-3}$ and 3.9 ng m$^{-3}$. $M_{rBC}$ showed a marked decrease below 500 m asl down to less than 1 ng m$^{-3}$. The size distribution of rBC also showed a clear altitude variability (Figure 2b), with the geometric mean of the mass size distribution increasing from approximately 180 - 190 nm aloft to 220 - 250 nm in the lowest 500 m. While the diameter
of rBC particles was reported to slightly decrease with altitude in summer in various Arctic regions (Jurányi et al., under review), the presence of large rBC particles in the lowest atmospheric layer is unusual for summer conditions (Arctic Ocean; Taketani et al., 2016). It must be considered that cloudy conditions dominated the ACLOUD campaign period, with cloud cover exceeding 70%, and preponderant occurrence of low-level clouds (Wendisch et al., 2018). Besides cloud layers observed at mid altitudes (2500 m asl), higher cloud droplet concentration was observed below 1000 m asl, with maximum values below
500 m asl (Figure 2c). The presence of low-level clouds was confirmed by collocated remote sensing measurements of the cloud top height performed on board of the Polar 5 aircraft with the AMALI lidar. The resulting vertical distribution of cloud top height frequency (Figure 2d) showed that 63% of observed clouds extended below 1000 m asl, with most of clouds (40% of the total) observed below 500 m.

Due to the variation of BC properties in correspondence of cloudy layers in the lowest 500 m of altitude; in the following, we
investigate the relationship between black carbon properties and low-level clouds to understand the potential impact of cloud processing on the vertical variability of BC properties in the summer Arctic boundary layer.

## 3.2   Identification of low-level, mixed-phase cloud cases

From 02 to 08 June, stable atmospheric conditions characterized by intrusion of warm and moist airmasses, absence of high-altitude clouds, and presence of low-altitude clouds were observed over sea ice north-west of Svalbard (Wendisch et al., 2018).
In this section, we will present the results obtained during four consecutive flights (02 June, 04 June, 05 June and 08 June), conducted north-west of Svalbard between approximately 80°N and 82°N (Figure 1b), which allowed investigating the variability of aerosol particles above and below cloud layers and, the relationship between cloud microphysics and rBC residuals.





Considering that the dynamic of the boundary layer strongly affects the properties, vertical extent and life-time of low-level clouds, we first present a meteorological characterization of the boundary layer from 02 to 08 June. It must be kept in mind that the vertical profiles presented in this section are an average of multiple ascents and descents of Polar 6, and that the cloud boundaries might have evolved in this time period. The boundary layer was well defined by a marked inversion, similar to Brooks et al. (2017), located at 400 - 500 m asl, where potential temperature ($T_P$) increased by approximately 269-273 K to approximately 277-282 K (Figure 3a). The cumulative mean of $T_P$ from the cloud base to the lowest sub-cloud measurement

was compared with $T_P$ calculated for each layer below-cloud. In general, the $T_P$ difference never exceeded 0.5 K, suggesting the presence of surface-coupled clouds (Gierens et al., 2020) and the establishment of a well-mixed boundary layer (Shupe et al., 2013). A moist atmospheric layer (RH > 80%) was confined by the inversion with values up to 98-100% between approximately 200 m asl and 500 m asl, while drier air (20% < RH < 50%) was observed above 500 m asl (Figure 3b). Within this moist boundary layer, liquid droplets were observed from a maximum altitude of 500 m asl to a minimum altitude of 50-

60 m asl (Figure 3c). Following the work of Sedlar et al. (2011), these cloud cases can be classified as cloud inside inversion, since the cloud top height extended above inversion-base height but below inversion-top height. Ice crystals with diameter larger than 75 μm measured with the CIP probe were observed from top of inversion to the lowermost altitude below cloud bottom, indicating the presence of mixed phase clouds and, ice sedimentation below cloud bottom during all selected flights (Figure 3d). Overall, the vertical atmospheric structure remained quite similar from 02 to 08 June, confirming the establishment

of stable atmospheric conditions maintaining mixed phase clouds in the boundary layer, as expected in Arctic summer conditions (Morrison et al., 2012).

Valid measurements of cloud residuals ranged from a maximum altitude of 544 m asl on 05 June to a minimum altitude of 60 m asl on 04 June. Overall, the vertical thickness of residual measurements varied from 310 m to 435 m, accounting for a total of 197 minutes (Figure S2). Above-cloud observations of aerosol particles were performed above the cloud-top and inversion

layer in the 400-750 m altitude range and are considered to be representative of free-tropospheric conditions. The vertical range covering the below-cloud bottom was very thin (100 - 150 m) and is considered to be representative of Arctic boundary layer impacted by mixed-phase clouds. The aerosol particle measurement time above cloud and below cloud was of 82 and 84 minutes, respectively; while total in cloud measurement time totalized 197 measurement minutes (Figure S2). The altitude range of in-, above-, and below- cloud measurements is shown in Figure 3e.

Dominant clear sky conditions were only observed on 25 June over sea ice north of Svalbard (Figure 1b). This flight was characterized by dry air (RH = 40 - 45%) above inversion (450 - 500 m asl) and a relatively moist boundary layer below (RH = 65 - 75%). No cloud droplets nor ice crystals were observed during the flight. The flight transects between 550 - 800 m asl were assumed to be representative of free tropospheric conditions, while the measurements performed below 200 m asl were considered to be representative of boundary layer conditions. In total, boundary layer observations lasted 170 minutes, while

free tropospheric conditions were measured for only 13 minutes (Figure S2).





### 3.3    rBC properties change from above to below cloud layer

The main objective of this section is to study in more detail the change of rBC properties already shown in in Figure 2 in the lowest atmospheric layers. Hence, we will present the variability of concentration and size distribution of rBC particles from above-cloud to below-cloud for the cloud events observed between 02 and 08 June, and for the clear sky event across inversion
on 25 June.

The flight ensemble median rBC mass concentration above cloud was 5.5 ng m$^{-3}$, with an interquartile range of 3.2 - 8.4 ng m$^{-3}$, these values are representative for Arctic pristine background and similar to previous Arctic summer concentration of rBC observed in free-troposphere (Schwarz et al., 2013; Roiger et al., 2015; Schulz et al., 2019). The median $M_{rBC}$ drastically decreased below clouds to 1.4 ng m$^{-3}$, with an interquartile range of 0.5 - 4.3 ng m$^{-3}$, similar to previous surface observations
of rBC particles over the Arctic Ocean in summer (Taketani et al., 2016). Overall, rBC mass concentration differed by a factor of 4 between above and below clouds, while rBC and total aerosol number concentration changed by approximately a factor of 3.7 and 1.7, respectively. Moreover, a $M_{rBC}$ decrease (36%) from 3.1 ng m$^{-3}$ in the free troposphere to 2.1 ng m$^{-3}$ in the boundary layer was observed in clear-sky condition on 25 June. The reduction of $M_{rBC}$ from free troposphere to boundary layer observed between 02 June and 08 June reflects the overall vertical profile of rBC mass showed in Figure 2, and might be
caused by both large scale transport (Croft et al., 2016), atmospheric stratification (Kupiszewski et al., 2013) and by cloud and sub-cloud processing (Gogoi et al., 2018).

The mass size distribution of rBC above-cloud was distributed around a mode at 177 nm for the event ensemble (Figure 4a). The distribution mode was fairly constant for all considered events ranging between 168 nm (02 June) and 189 nm (08 June) with an overall geometric mean of 189 nm. Similar values were observed in the free troposphere under clear-sky condition on
25 June (Figure 4d), when a mode at 183 nm and a geometric mean of 190 nm were observed. These observations indicated the establishment of stable conditions in the free troposphere, independent from cloud presence below inversion-top and similar to previous Arctic summer (Schulz et al., 2019) and spring (Raatikainen et al., 2015; Zanatta et al., 2018) observations. Below-cloud, the size distribution of rBC aerosol was remarkably different (Figure 4b), and characterized by the presence of a mode below 200 nm and a shoulder peaking in a prominent overflow bin including all rBC particles larger than the upper
quantification limit ($D_{rBC} > 575$ nm). The recurrent presence of a peak around 180 nm below and above clouds suggests that bulk of the rBC aerosol population had a similar origin and underwent similar ageing processes. However, the presence of particles larger than 300 nm caused a shift of the geometric size distribution from 189 nm above-cloud to 251 nm below-cloud. None of the Arctic studies based on SP2 measurements ever reported a similar rBC size distribution, neither in the boundary layer nor in the free troposphere in summer or spring (Raatikainen et al., 2015; Taketani et al., 2016; Kodros et al., 2018;
Zanatta et al., 2018; Schulz et al., 2019; Ohata et al., 2021). The rBC size distribution observed in clear-sky conditions (25 June) within the boundary layer did not show a clear mode in the SP2 size detection range and was depleted in particles larger than 150 - 200 nm (Figure 4d).





All told, these observations confirmed the general vertical variability presented in Section 3.1, clearly showing that ground observations are not representative of the free troposphere. Moreover, considering the altitude depended variation of the rBC

size distribution with respect to cloud layer, we hypothesize that cloud processing might be responsible for the change of rBC size from above to below the cloud layer.

### 3.4    Cloud activation of rBC particles

To understand if cloud processing played a role in the variation of rBC properties across the cloud layers, we further

investigated the microphysical properties of cloud particles and of rBC cloud residuals. First, we characterized the general microphysical properties of the selected clouds to assess their representativity of Arctic summer conditions. Measurements of cloud residuals occurred in relatively warm conditions (temperature interquartile range of IQR -5.8 - -3.9°C), as expected in summer for low-level and mixed-phase clouds in the Arctic Ocean (Achtert et al., 2020). In this temperature regime, $N_{Dro}$ dominated the concentration of cloud particles with a cloud ensemble median concentration of 75.8 cm$^{-3}$, while $N_{Ice}$ was 2.65

L$^{-1}$. As a consequence, liquid water dominated the cloud phase with a median LWC of 0.15 g cm$^{-3}$ (IQR = 0.095 - 0.21 g cm$^{-3}$) leading to a median IWF of 1.1% (calculated as the ratio of IWC and TWC; Section 0). Griesche et al. (2021) confirmed, by means of remote observation during the collocated PASCAL experiment (Flores and Macke, 2018), the minor presence of ice crystals at temperature above -10°C. The droplet and ice crystal properties observed in this work for the cloud ensemble were comparable to previous observations performed in the European Arctic between April and May (Mioche et al., 2017) and

in the Arctic Ocean between July and August (Achtert et al., 2020). Combining these observations with results of Section 3.2, the cloud events discussed here show many features common to Arctic persistent mixed-phase clouds as summarized by Sedlar et al. (2011), Morrison et al. (2012) and Korolev et al. (2017): dominance of supercooled droplets, intruding-inversion clouds but coupled with the surface and ice sedimentation below-cloud. We can thus conclude that the selected ACLOUD cloud cases were not extreme events and fairly represented summer Arctic conditions.

Second, we proceeded to compare the properties of rBC residuals such as concentration, size distribution and mixing state with rBC particles sampled above and below cloud.

### 3.4.1    Concentration of rBC residuals

Under this cloud regime, the number concentration of rBC particles measured behind the CVI inlet in cloud was low, with median $N_{rBC-res}$ of 0.59 cm$^{-3}$ and interquartile range of 0.30 - 1.0 cm$^{-3}$. As a result, only a minor number of cloud droplets

contained an rBC particle. The $N_{rBC-res}/N_{Dro}$ ratio ranged from a maximum median of 1.5% on 02 June to a minimum median of 0.7% on 04 June, with a cloud ensemble median of 0.91% and interquartile range of 0.47 - 1.4% (Figure 5a). Although these results clearly indicated that rBC particles are a minority of cloud active aerosol particles, to assess the activated fraction of rBC particles, we compared the concentration of rBC residuals with $N_{rBC}$ sampled above and below clouds. This approach was already used in the Alaskan Arctic in spring in presence of liquid clouds by Earle et al. (2011), who calculated the ratio between





the number concentration of liquid droplets ($N_\text{Dro}$) and the number concentration of aerosol particles sampled below cloud ($N_\text{AP-blw}$) to address the aerosol activation process. During the ACLOUD cases, the median $N_\text{Dro}/N_\text{AP-blw}$ ratio was 1.1 with interquartile range of 0.76 - 1.3 suggesting that the totality of below-cloud aerosol particles was activated in-cloud. It must be noted that $N_\text{Dro}/N_\text{AP-blw}$ need to be interpreted carefully. Due to the lower quantification size limit of the UHSAS (80 nm of optical diameter), $N_\text{AP}$ and $N_\text{AP-blw}$ do not include the smaller aerosol particles; hence, the $N_\text{Dro}/N_\text{AP-blw}$ ratio presented here is

inevitably overestimated with respect to the total aerosol particle population. Our results nicely agree with Earle et al. (2011), who presented an average $N_\text{Dro}/N_\text{AP-blw}$ value of 0.99, and with McFarquhar et al. (2011), who showed an almost 1:1 relationship between $N_\text{Dro}$ and $N_\text{AP-blw}$ in Arctic pristine conditions. We then proceeded to calculate the ratio between $N_\text{rBC-res}$ over $N_\text{rBC}$ measured below-cloud ($N_\text{rBC-blw}$) and above-cloud ($N_\text{rBC-abv}$). $N_\text{rBC-res}/N_\text{rBC-abv}$ varied from the highest median value of 0.44 on 02 June to the lowest median value of 0.37 on 08 June, with a cloud ensemble median of 0.31 (IQR=0.18-0.47; Figure 5b).

The $N_\text{rBC-res}/N_\text{rBC-blw}$ median values were surprisingly high, ranging from a minimum of 1.05 on 02 June and a maximum of 1.34 on 04 June, with a cloud ensemble median of 1.17 and interquartile range of 0.72-1.76 (Figure 5c). The above unity values of both $N_\text{Dro}/N_\text{AP-blw}$ and $N_\text{rBC-res}/N_\text{rBC-blw}$ indicated that the totality of both aerosol particles and rBC particles below cloud be activated in cloud (McFarquhar et al., 2011) via adiabatic lifting (see process 1 in Figure 6a; Earle et al., 2011), but also that other Arctic relevant activation mechanisms might occur. Aerosol particles might be entrained from above-cloud to directly

promote droplet formation (see process 2 - 3 in Figure 6a; Igel et al., 2017) or being scavenged from the interstitial phase via coagulation with pre-existing droplets (see process 4 in Figure 6a; Croft et al., 2016). These two mechanisms might enhance the number of activated rBC particles, potentially explaining the above unity values of both $N_\text{Dro}/N_\text{AP-blw}$ and $N_\text{rBC-res}/N_\text{rBC-blw}$. Clearly, the contribution of measurement uncertainty to values above unity of $N_\text{rBC-res}/N_\text{rBC-blw}$ must be considered. The uncertainty of $N_\text{rBC-res}$ has multiple contributions: the 1σ reproducibility of $N_\text{rBC}$ measured by SP2 (5%; Laborde et al., 2012),

and the uncertainty associated with the transmission efficiency factor (13%). The latter was estimated by propagating the 1σ reproducibility associated with $N_\text{AP}$ measured by the UHSAS (9%; Ehrlich et al., 2019) and the uncertainty of $N_\text{Dro}$ measured by the SID-3 (10%; Baumgardner et al., 2017).

). The overall uncertainty of $N_\text{rBC-res}/N_\text{rBC-blw}$ was finally estimated at 15%. To better understand the contribution of below-cloud activation or entrainment from the free troposphere of rBC particles we further compared the size distribution and mixing

state of rBC residuals with above- and below-cloud rBC particles.

### 3.4.2 Size distribution of rBC residuals

The mass size distribution of rBC residuals was reasonably similar during all cloud cases (Figure 4c), indicating similar conditions along the measuring period. The mass size distribution of the rBC residuals showed similar features to those of rBC particles sampled both above and below cloud. First, a mode at 193 nm was evident in the rBC residuals size distribution,

being comparable to rBC particles observed above and below cloud. Second, rBC residuals showed a prominent shoulder towards larger diameters, culminating in the overflow saturation bin (rBC cores larger than 575 nm). This feature was shared only with rBC particles sampled below-cloud. To quantify the size-dependent enrichment or depletion of rBC in cloud residuals





compared to outside cloud, we calculated the ratio of the number size distribution of rBC residuals over the number size distribution of rBC particles sampled above-cloud and below-cloud. Figure 7a show that larger rBC particles were enriched in cloud residuals compared to above-cloud conditions. The ratio of the size distribution monotonically increased from 0.3 in the 80-100 nm diameter range to values between 1.2 and 3.5 in the 440 - 575 nm diameter range. The ratio of the number size distribution was different for below-cloud rBC (Figure 7b). First, rBC particles were more numerous in cloud residuals compared to below-cloud at all diameters, with cloud-ensemble median values near or above unity for all $D_{rBC}$. Similar to the result presented in Section 3.4.1 and in Figure 5. the ratio increased from approximately 1 for $D_{rBC}$ below 100 nm to values around 1.5 for $D_{rBC}$ larger than 200 nm, indicating that, although larger particles were still preferentially observed in the cloud phase rather than below cloud, the size segregated population of rBC particles was fairly similar between cloud phase and below cloud conditions. The variability of the size distribution across the cloud layers supports the hypothesis of concurring activation mechanisms: above cloud entrainment might be responsible for the recurring presence (above, in and below cloud) of rBC particles with diameters around 200 nm, while the rBC particles larger than 300 nm might be preferentially activated from below cloud.

### 3.4.3 Coating thickness of rBC residuals on 02 June

Due to failure of the scattering detector, the quantification of coating thickness was only possible for the flight occurred on 02 June. Considering that the coating thickness was quantified for rBC cores in the 200 - 300 nm diameter range, which represented a small subset of the total detected rBC particles, the results discussed as following are extremely uncertain due to the low counting statistics. First, the mixing degree of rBC cores in the 200 - 250 nm range will be discussed. The thinnest coatings were observed above clouds, where the coating thickness median was 30 nm (IQR = 23 - 48 nm) and median shell-to-core ratio was 1.51 (IQR = 1.38 - 1.8). rBC cloud residuals showed slightly thicker coatings, with a median value of 38 nm (IQR = 25 - 59 nm) and median shell-to-core ratio of 1.58 (IQR = 1.39 – 1.92 nm). The thickest coatings were observed below clouds, where the median coating thickness was 43 nm (IQR = 25 - 58 nm), and median shell-to-core diameter ratio was 1.67 (IQR = 1.43 - 1.98). A similar increasing trend from above-cloud (median thickness 22 nm, median shell-to-core diameter ratio of 1.37) in-cloud (median thickness 27 nm, median shell-to-core diameter ratio of 1.41) and below-cloud (median thickness 32 nm median shell-to-core diameter ratio of 1.60) was also observed for larger rBC cores (250 - 300 nm). Although no estimations of the coating thickness are available for the Arctic region in summer and for cloud residuals, our results are in a similar range compared with previous Arctic spring observations at ground sites (Raatikainen et al., 2015; Zanatta et al., 2018) and airborne (Kodros et al., 2018; Ohata et al., 2021) where shell-to-core varied from 1.5 to 1.7. Like the increase of mean diameter, the increase of coating thickness and shell-to-core ratio from above to below clouds might suggest that cloud active rBC and below-cloud rBC underwent similar ageing process, potentially driven by cloud activation.





### 3.5 Vertical structure of cloud microphysics and rBC residuals

In the following, we addressed the vertical structure of cloud phase to identify any cloud processes and potential effects on rBC residuals properties. Due to the low counting statistics caused by the low concentration of ice crystals and rBC particles, this analysis is based exclusively on the cloud ensemble. Considering that cloud top and cloud bottom height showed some variability during the different flight (Figure 3), the in-cloud normalized altitude ($Z_n$) was calculated following Mioche et al. (2017) as:

$$Z_n = \frac{Z - Z_b}{Z_t - Z_b} \qquad\qquad 2$$

Where $Z$ is the measurement altitude, $Z_b$ is the lowest altitude of in-cloud valid rBC-residuals measurements and $Z_t$ is highest altitude of in-cloud rBC-residuals measurements. Thus, $Z_n = 1$ and $Z_n = 0$ correspond to the highest and lowest rBC-residuals measurement, respectively. Considering the relatively thin clouds (vertical extent between 310 - 435 m) compared to Mioche et al. (2017), the cloud was divided into only 5 layers. Liquid droplet number concentration varied between 58 cm$^{-3}$ and 81 cm$^{-3}$ with a clear maximum in the middle of the cloud (Figure 8a), while the increase of $D_{Dro}$ and LWC (Figure 8b and Figure 8c)

from cloud bottom to cloud top agrees with cloud observations in the European (Mioche et al., 2017) and Alaskan (Earle et al., 2011) Arctic in spring. It is worth noticing that a slight decrease of LWC (as a product of $N_{Dro}$ and $D_{Dro}^3$) and, increase of temperature (approximately 1°C, Figure S3a) was observed in the top fifth of the cloud layer. This feature might be associated with the entrainment of warmer and dryer air from the inversion layer (Sedlar et al., 2011; Zhao et al., 2018) and corroborate the hypothesis of entrainment and subsequent activation of smaller rBC particles from the free troposphere, (see process 2 in

Figure 6a; Igel et al., 2017).

The ice phase showed an opposite vertical trend. From cloud top to cloud bottom, the median $N_{Ice}$ increased by one order of magnitude (from 1.1 L$^{-1}$ to 7.2 L$^{-1}$; Figure 8d), $D_{Ice}$ increased from 167 µm to 278 µm (Figure 8e), and IWC increased from 0 g m$^{-3}$ to 0.018 g m$^{-3}$ (Figure 8f). As a consequence of the inverse proportionality between LWC and IWC, the IWF increased from 0% at cloud top to 18% at cloud bottom (Figure S3b). Considering the lowest size detection limit of the CIP instrument

(75 µm), the vertical profiles of ice crystal properties might be biased. Nonetheless, similar vertical variability of the liquid phase in mixed-phase cloud was already observed in spring and summer in the European Arctic and in the Arctic Ocean, (Jackson et al., 2012; Mioche et al., 2017). Increase of IWC might take place via collection of multiple liquid droplets by large falling ice crystals or under turbulent conditions (Grabowski and Wang, 2013). During the ACLOUD campaign, the majority of ice crystals with diameters between 20 µm and 700 µm identified by the PHIPS probe showed riming (30-47%; Figure S3c

and Waitz et al., 2022). The occurrence of other processes such as contact freezing, rime-splintering or ice aggregation was not clear from the PHIPS data, which showed absence of aggregate, frozen droplets and graupels.

Under this very complex cloud condition, rBC residuals number concentration showed a maximum in the two lowest cloud layers (Figure 8g), where the median $N_{rBC\text{-}res}$ varied between 0.67 cm$^{-3}$ and 0.78 cm$^{-3}$. $N_{rBC\text{-}res}$ decreased monotonically in the cloud layers aloft, reaching the lowest median values at cloud top (0.46 cm$^{-3}$). The relative difference of $N_{rBC\text{-}res}$ from cloud

bottom to cloud top was of -30%, well above the overall uncertainty associated with $N_{rBC\text{-}res}$ (14%, contribution of SP2 and





transmission efficiency factor), while the layer-by-layer relative change of $N_{rBC-res}$ (7% - 21%) was above the 1σ reproducibility of $N_{rBC}$ (5%; Laborde et al., 2012) but in the range of $N_{rBC-res}$ uncertainty. The mass size distribution of the rBC residual population also showed a vertical trend, with larger particles enriched at cloud bottom, where a median diameter of 334 nm was observed. $D_{rBC-res}$ decreased in the cloud layers above, where median values varied between a minimum and maximum

range of 222 - 253 nm (Figure 8h). In general, $D_{rBC-res}$ decreased by 33% from cloud bottom to cloud top, this decrease being larger than the overall uncertainty associated with $M_{rBC-res}$ (17%, contribution of SP2 and transmission efficiency factor). The statistical significance of $N_{rBC-res}$ and $D_{rBC-res}$ vertical trend was verified with T-test analysis. T-test was applied to sub-datasets representing adjacent vertical layers, from the bottom to the top of the cloud. The hypothesis of equal averages, for both number concentration and diameter of rBC particles is respected only for the first two layers ($Z_n$ below 0.4), indicating that no trend

was statistically significant across the two lowermost cloud layers. Otherwise, the layer-by-layer difference was statistically significant for the upper cloud layers ($Z_n$ above 0.4). Hence, the rBC residual population appeared to be significantly different, at least, at cloud bottom and cloud top. These results show a very defined stratification of cloud phase and properties of rBC residuals, with a dominant liquid phase and smaller rBC residuals at cloud top and enriched ice phase and larger rBC residuals at cloud bottom.


## 4    Discussion on the potential cloud processing mechanisms affecting the vertical distribution of rBC particles

The aim of this section is to present all mechanisms which might have contributed to the shift in size and mixing of rBC particles from the free troposphere to the cloud dominated boundary layer. Moreover, the interpretation of the results presented in this works is particularly complicated by the role of the local atmospheric dynamic such as updraft, downdraft and

turbulence, which play a key role by redistributing aerosol, water vapor and hydrometeors. Considering that we could not prove nor discard with absolute certainty the occurrence of any of the proposed mechanisms, this section should be considered merely, as an open discussion and regarded with caution.

Microphysical properties of clouds are tightly connected with aerosol load through various indirect effects. Among other mechanisms, the first aerosol indirect effect, also known as Twomey effect, describes the decrease of cloud droplet effective

radius ($Re_{Dro}$) and increase of cloud droplet number concentration caused by the increase of CCN load under constant or limited range of LWC (Twomey, 1977). Although the Twomey effect was already observed in the Arctic (Garrett and Zhao, 2006; Mauritsen et al., 2011; Coopman et al., 2016, 2018), little is known on its relationship with CCN diameter and presence of ice phase. The occurrence of the Twomey effect was investigated in two separate cloud layers with distinct LWC values. The highest LWC values were observed above cloud-mid ($0.6 < Z_n < 0.8$), where LWC values varied between 0.12 and 0.25 g m$^{-3}$

(interquartile range). $N_{Dro}$ showed a negative proportionality with $Re_{Dro}$ in this mid-layer of the clouds, (Figure 9), where the median temperature was -6°C and the IWF 75[th] percentile was below 3%. Within this cloud regime, hygroscopic rBC particles with a geometric mean diameter around 220 nm (Figure 8h) can efficiently act as CCN, contributing, even if in small number





(median $N_{\text{rBC-res}}/N_{\text{Dro}} = 0.7\%$) to maintain the Twomey effect, by both activation from the sub-cloud layer followed by adiabatic

lifting (see process 1 in Figure 6a; Earle et al., 2011) or by above-cloud entrainment (see process 2 - 3 in Figure 6a; Igel et al.,

2017). The lowermost LWC regime (LWC values with a median of 0.085 g m$^{-3}$ and an interquartile range of 0.062 - 0.10 g m$^{-3}$) was observed at cloud bottom ($0.0 < Z_{\text{n}} < 0.2$), where the median temperature was warmer ($T$ median = -4.2°C) and the

presence of ice was remarkably higher than the high-LWC regime. The interquartile range of the IWC was 0.007 - 0.04 g m$^{-3}$,

associated with an increase of the ice water fraction up to a median value of 21% and a 75$^{\text{th}}$ percentile value of 34%. The

proportionality between Re$_{\text{Dro}}$ and $N_{\text{rBC-res}}$ was substantially different from high-LWC regime and characterized by a positive

correlation (Figure 9), suggesting the occurrence of alternative aerosol-cloud interaction. This positive correlation was already

observed in warm clouds and alternatively called "anti-Twomey" or "reverse Twomey" effect (Jose et al., 2020; Jose, 2022).

Although largely unknown, Twomey effect could be reversed by the activation of large CCNs or excessive presence of CCNs

in marine environment (Jose et al., 2020; Jose, 2022) or by droplet coalescence in presence of drizzle (Zhao et al., 2018) or in

extremely polluted environments (Khatri et al., 2022). This told, the increase of rBC residuals activated fraction (median $N_{\text{rBC-res}}/N_{\text{Dro}} = 1.2\%$) and size (geometric mean of $D_{\text{rBC-res}} \approx 330$ nm; Figure 8h) compared to high-LWC regime might not be caused

only by direct activation of hygroscopic rBC particles, but also by processes competing with the Twomey effect. We first will

present the in-cloud processes that might potentially compete with Twomey and lead to the size growth of rBC particles

contained in the liquid and ice phase (Figure 6b). Second, the mechanism responsible for transfer of rBC at cloud interphase

will be presented (Figure 6c). Droplet coalescence or coagulation is known to promote the growth of liquid droplets and the

decrease of their absolute number concentration, thus reversing the Twomey effect (Zhao et al., 2018). The coalescence of two

or more droplets, each containing a single rBC particles, would also increase above unity the number of cloud active rBC

inside the same droplet (see process 1 in Figure 6b; Ding et al., 2019). Upon evaporation in the CVI, a larger single rBC

particle would then be released and detected by the SP2. As also shown by Ding et al. (2019), this process might explain the

presence of large rBC residuals in the cloud phase, otherwise not observed in the free troposphere. Considering the low

collision probability of two droplets containing an rBC particle (approximately 1% of droplet containing rBC particle), and

that the in-cloud vertical profiles show an anticorrelation between $D_{\text{Dro}}$ and $D_{\text{rBC}}$ (Figure 8); droplet coalescence might not be

considered an efficient growth mechanism for rBC residuals in the ACLOUD cases. Icing processes such as riming might

actually compete with the Twomey effect (Lohmann and Feichter, 2005; Jackson et al., 2012) and also cause a change in the

size of rBC particles and its partitioning between the liquid, ice and interstitial phase (Ding et al., 2019). As already shown in

Figure 8 and in Figure 9, the ice phase ($N_{\text{Ice}}$, $D_{\text{Ice}}$, IWC) increased at the expenses of liquid water especially in the lowermost

part of the cloud, while most of the ice crystals showed rimed habit (Figure S3). We thus argued that riming might have

occurred during the cloud cases considered in this study. The capture of several liquid droplets by a precipitating ice crystal,

might cause the presence of multiple CCN particles in a single precipitating ice crystal, which would be released as a larger

rBC aggregate (see process 2 in Figure 6b; Ding et al., 2019). However, the question "how droplet coalescence and riming

might be connected to the increase of the diameter of rBC particles sampled below cloud?" arises. rBC residuals can be released

at cloud bottom or below-cloud via three mechanisms (Figure 6c). First, water vapor deposition from liquid droplets to ice





crystals (WBF) would contribute to the decrease of LWC in the lowest cloud layers, and, upon complete consumption of liquid droplets, would release the former rBC residuals to the interstitial phase at cloud-bottom interphase (see process 1 in Figure 6c; Qi et al., 2017; Ding et al., 2019). Alternatively, the large and rimed ice crystals would precipitate and release rBC

aggregates upon sublimation below cloud (see process 2 in Figure 6c; Solomon et al., 2015). Similarly, due to sedimentation of large cloud droplets and sub-cloud evaporation, large rBC aggregates can be released below cloud bottom (see process 3 in Figure 6; Igel et al., 2017). It is, however, unclear if atmospheric conditions below cloud such as humidity, temperature and windspeed might efficiently favour evaporation over sublimation, or if activated rBC particles would simply be removed due to precipitation. Solomon et al. (2015) showed that recycling of ice nucleating particles via sedimentation, sublimation and

reactivation contributes to maintain the ice phase in mixed-phase clouds over several days in the Arctic region (see process 4 in Figure 6c). Considering that rBC ice activity in warm temperatures is negligible (Kanji et al., 2020), the large and internally mixed rBC particles released by WBF at cloud bottom or by sublimation of rimed crystals or by evaporation of liquid droplets below cloud might be reactivated as CCN. In case of persistent Arctic clouds, rBC particles could undergo an activation-growth-release cycle multiple time; thus, the rBC metamorphism presented in the result section would be results of several

iterations.

## 5    Conclusion

The interaction of black carbon particles with Arctic clouds was investigated with airborne measurements in the north west of Svalbard (Norway) in the framework of the ACLOUD campaign in summer 2017. The overall vertical variability of rBC properties during the ACLOUD campaign indicated a net decrease of rBC concentration and increase of rBC diameter in the

lowest atmospheric layer dominated by clouds. Four case events characterized by the presence of low-level, surface-coupled, inside-inversion and mixed-phase clouds were identified.

The analysis of these events confirmed a net separation of rBC properties from the atmospheric layers above-cloud in the free troposphere to the below-cloud layer, where less (median rBC mass concentration of 1.4 ng m$^{-3}$), larger (geometric mean of the mass size distribution of 251 nm) and more coated (median coating thickness of 43 nm) rBC particles were observed

compared to above-cloud conditions (median rBC mass concentration of 5.5 ng m$^{-3}$, geometric mean of the mass size distribution of 189 nm, and median coating thickness of 30 nm). Interestingly, in the absence of clouds, rBC particles in the boundary layer were dominated by small diameters (upper limit of geometric mean of the mass size distribution of 147 nm). Under mixed-phase cloud conditions (median temperature of -5.3°C, median LWC of 0.15 g cm$^{-3}$, and median IWC of 0.002 g cm$^{-3}$), only a small minority of droplets (less than 1%) contained an rBC particle. However, it appeared that the totality of

rBC particles below the cloud layer was activated, with potential additional contribution to activated fraction via entrainment of rBC particles from the free-troposphere. Interestingly, the population of rBC residuals was enriched in larger (geometric mean of the mass size distribution of 249 nm) and thickly coated (median coating thickness of 38 nm) particles, very similar





to below-cloud rBC. It thus appeared that the presence of larger particles in-cloud and below-cloud and their absence above-cloud might be connected with cloud processing.

Studying the vertical variability of cloud particles and rBC residuals, it became evident that the cloud top underwent completely different conditions compared to cloud bottom, were the increase of ice fraction was associated with a deviation from Twomey effect and potentially connected with cloud processing inducing the growth of rBC residuals. In view of our results we proposed that the large and coated particles observed below cloud are the results of a processing cycle including activation, cloud processing such as riming or droplet coalescence and sub-cloud release mechanism such as sublimation/WBF or evaporation,

followed by reactivation. Although we could not prove this hypothesis, in case of persistent low-level Arctic clouds this cycle might be repeated multiple times, making cloud processing a mechanism controlling the properties of black carbon in the boundary layer.

To conclude, the ACLOUD observations demonstrated that surface observations are clearly not representative of the atmosphere aloft. This statement becomes particularly important in presence of low-level mixed-phase, persistent clouds; when

cloud processes might influence not only the vertical distribution of black carbon but also its microphysical properties over long time periods.

*Code and data availability*

The SP2 data were analysed with PSI Toolkit single particle soot photometer (SP2), version 4.110. Contact Droplet

Measurement Technologies to download the software.

*Author contributions*

The manuscript was written by MZ with contributions from all authors. Cloud particle measurements and subsequent data analysis were performed by OJ, RD, EJ and MS. Aerosol particle measurements and subsequent data analysis were performed

by MZ AH, ZJ, SM, OE and JS. All authors contributed to the data analysis

Acknowledgements

The authors thank Christof Lupkes (Alfred-Wegener-Institut, Helmholtz-Zentrum für Polar- und Meeresforschung (AWI), Bremerhaven, Germany) and and Roland Neuber ([1]Alfred-Wegener-Institut, Helmholtz-Zentrum für Polar- und

Meeresforschung (AWI), Potsdam, Germany) for the valuable technical and scientific discussion.

*Financial support*

We gratefully acknowledge the funding by the Deutsche Forschungsgemeinschaft (DFG, German Research Foundation)– project ID 268020496 – TRR 172, within the Transregional Collaborative Research Center "ArctiC Amplification: Climate

Relevant Atmospheric and SurfaCe Processes, and Feedback Mechanisms (AC)3". Marco Zanatta acknowledges funding by the Deutsche Forschungsgemeinschaft (DFG, German Research Foundation, grant no. Projektnummer 457895178).



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



# Tables

**Table 1 List of atmospheric variables observed and computed in this study. Including meteorology, aerosol particle, cloud particle residuals and cloud particles.**

| Variable | Symbol | Unit | Instrument | Inlet | Aircraft | Size range (diameter) |
|---|---|---|---|---|---|---|
| | | | | | | |
| **Meteorology** | | | | | | |
| Temperature | $T$ | °C | - | - | P6 | - |
| Potential temperature | $T_P$ | °C | - | - | P6 | - |
| Relative humidity | RH | % | - | - | P6 | - |
| Pressure | $P$ | hPa | - | - | P6 | - |
| | | | | | | |
| **Aerosol particle** | | | | | | |
| rBC mass conc. | $M_{rBC}$ | ng m$^{-3}$ | SP2 | Total | P6 | 73-570 nm |
| rBC number conc. | $N_{rBC}$ | cm$^{-3}$ | SP2 | Total | P6 | 73-570 nm |
| rBC diameter | $D_{rBC}$ | nm | SP2 | Total | P6 | 73-570 nm |
| AP number conc. | $N_{AP}$ | cm$^{-3}$ | UHSAS | Total | P6 | 80-1000 nm |
| AP diameter | $D_{AP}$ | nm | UHSAS | Total | P6 | 80-1000 nm |
| | | | | | | |
| **Cloud particle residuals** | | | | | | |
| rBC number conc. | $N_{rBC\text{-}res}$ | cm$^{-3}$ | SP2 | CVI | P6 | 73-570 nm |
| rBC diameter | $D_{rBC\text{-}res}$ | nm | SP2 | CVI | P6 | 73-570 nm |
| AP number conc. | $N_{AP\text{-}res}$ | cm$^{-3}$ | UHSAS | CVI | P6 | 80-1000 nm |
| | | | | | | |
| **Cloud particles** | | | | | | |
| Droplet number conc. | $N_{Dro}$ | cm$^{-3}$ | SID-3 | - | P6 | 10-45 µm |
| Droplet diameter | $D_{Dro}$ | µm | SID-3 | - | P6 | 10-45 µm |
| Droplet effective radius | $Re_{Dro}$ | µm | SID-3 | - | P6 | 10-45 µm |
| Liquid water content | $LWC$ | g m$^{-3}$ | SID-3 | - | P6 | 10-45 µm |
| Ice crystal number conc. | $N_{Ice}$ | L$^{-1}$ | CIP | - | P6 | 75-1550 µm |
| Ice crystal diameter | $D_{Ice}$ | µm | CIP | - | P6 | 75-1550 µm |
| Ice water content | $IWC$ | g m$^{-3}$ | CIP | - | P6 | 75-1550 µm |
| Ice water fraction | $IWF$ | - | CIP - SID-3 | - | P6 | 10-1550 µm |
| Ice crystal habit | - | - | PHIPS | - | P6 | 20-700 µm |
| Clout top height | - | m asl | AMALI | - | P5 | - |
| | | | | | | |




# Figures

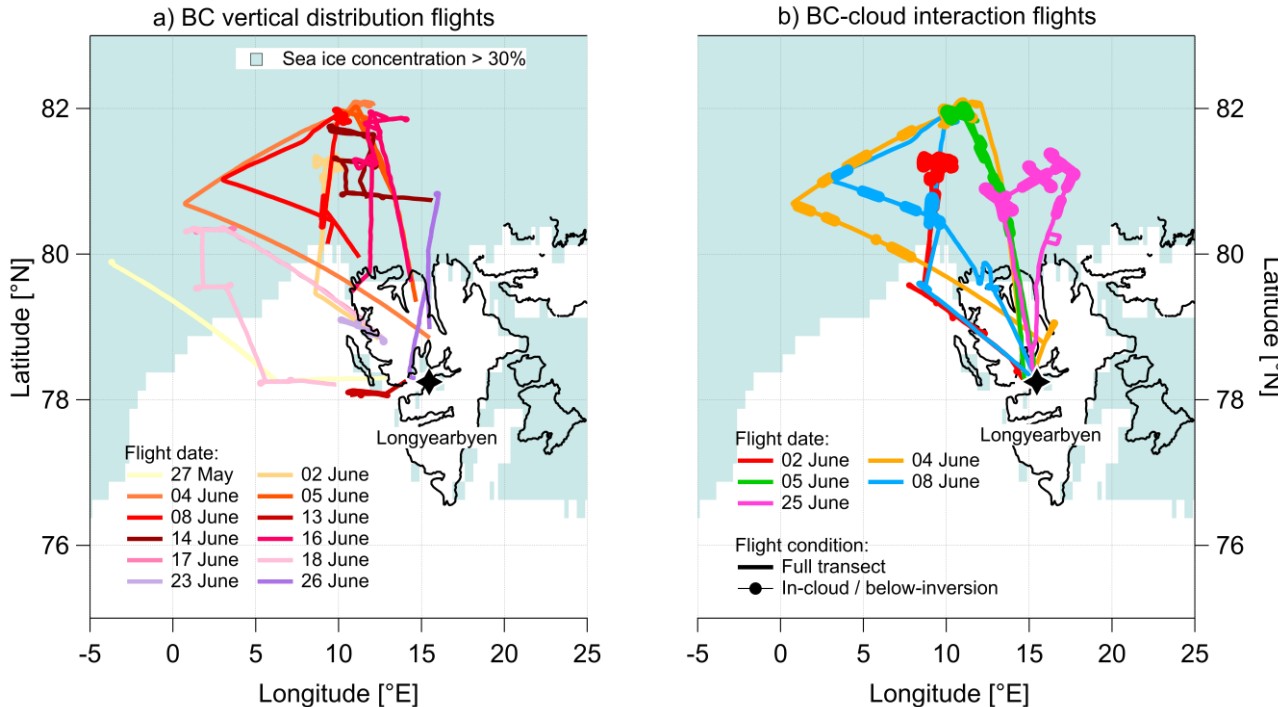

**Figure 1 Map of Svalbard including flight patterns of the Polar 6 aircraft for the flights dedicated to investigating the vertical distribution (a) and cloud interaction (b) of BC particles. Sea ice concentration derived from the GHRSST Sea Surface Temperature Level 4, MUR25 sea surface temperature analysis product with 0.25° resolution (MUR-JPL-L4-GLOB-v4.1 doi:10.5067/GHGMR-4FJ04).**






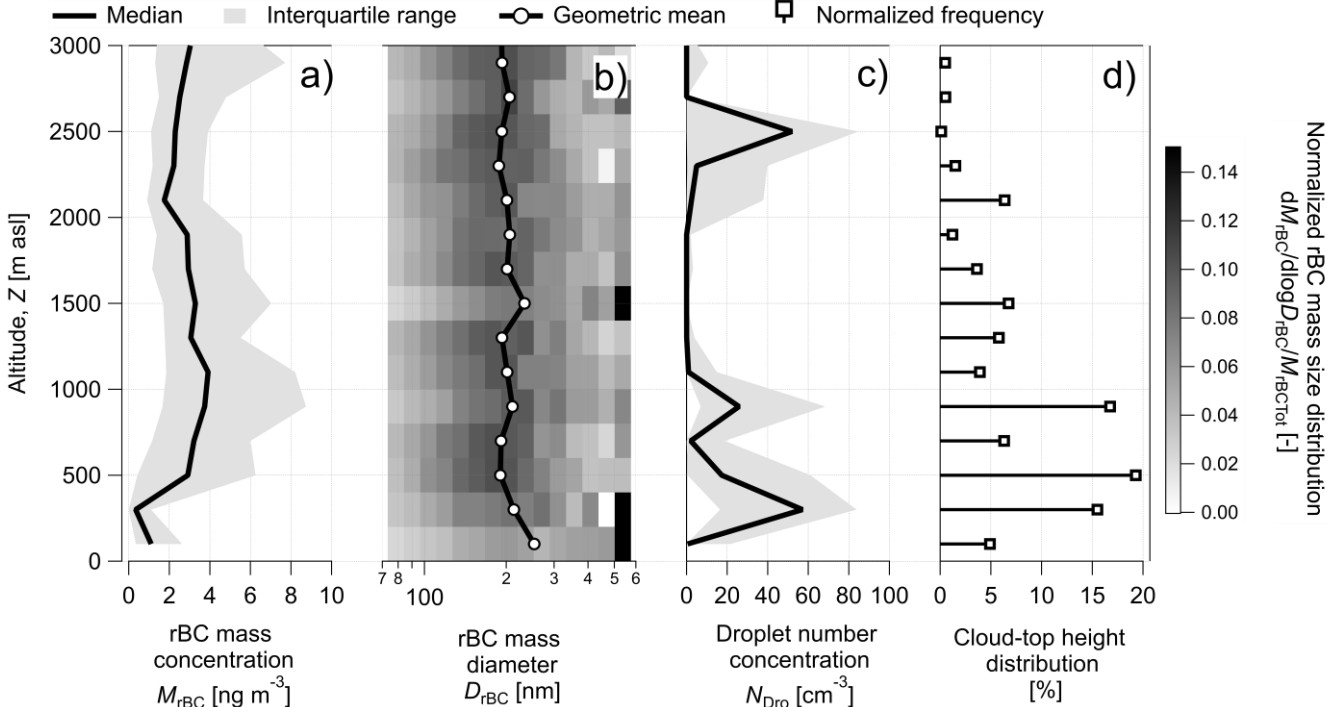


**Figure 2 Vertical variability of: a) rBC mass concentration; b) rBC mass size distribution; c) droplet number concentration; d) cloud top height. rBC particles sampled behind the aerosol inlet and measured with the SP2 in the 75-575 nm diameter range. Liquid droplets measured with the SID-3 probe in the 10-45 µm diameter range. Cloud top derived from the AMALI instrument. rBC particles sampled behind the aerosol inlet and measured with the SP2 in the 75-575 nm diameter range. Statistics calculated for**

**equidistant altitude steps starting at the surface (0m asl) and 100m thick.**





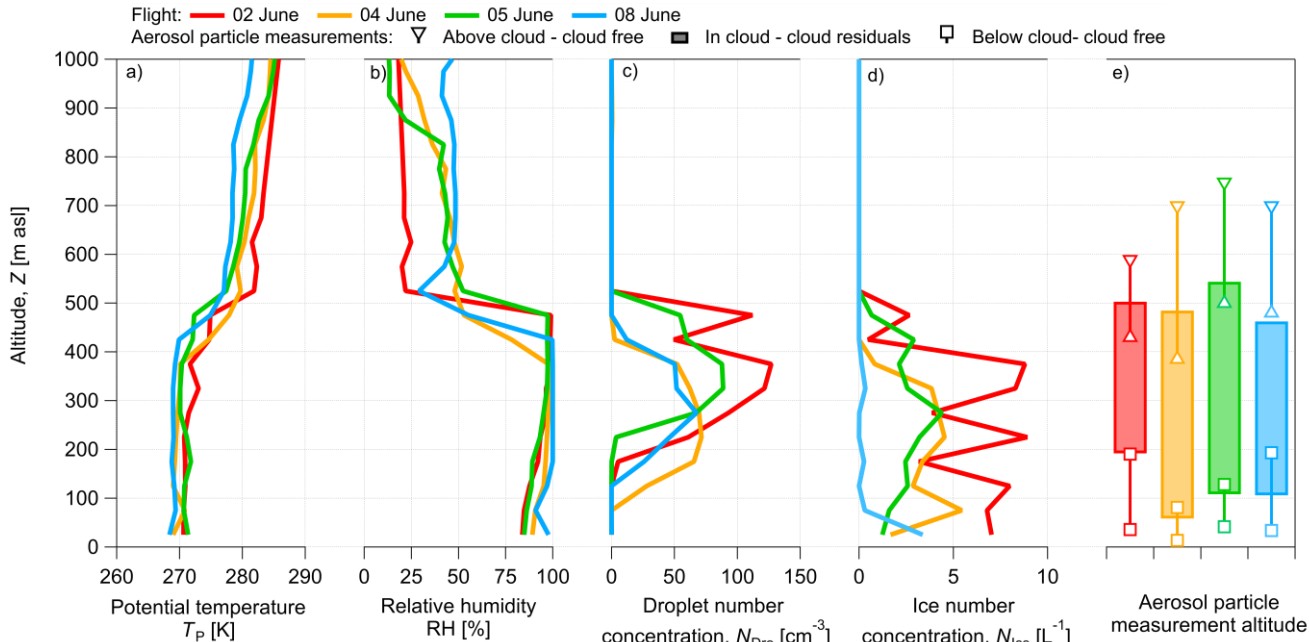

**Figure 3 Atmospheric and cloud characterisation during four flights occurred on 2, 4, 5 and 8 June 2017 north-west of Svalbard.**
**Median vertical profile of: a) potential temperature, TP; b) relative humidity, RH; c) number concentration of droplets, $N_{Dro}$; d)**
**number concentration of ice crystals, $N_{Ice}$; e) altitude range of aerosol particle measurement in different conditions. Liquid droplets**
**measured with the SID-3 probe in the 10-45 μm diameter range. Ice crystals measured with the CIP probe in the 75-1550 μm**
**diameter range. Statistics calculated for equidistant altitude steps starting at the surface (0m asl) and 50 m thick.**





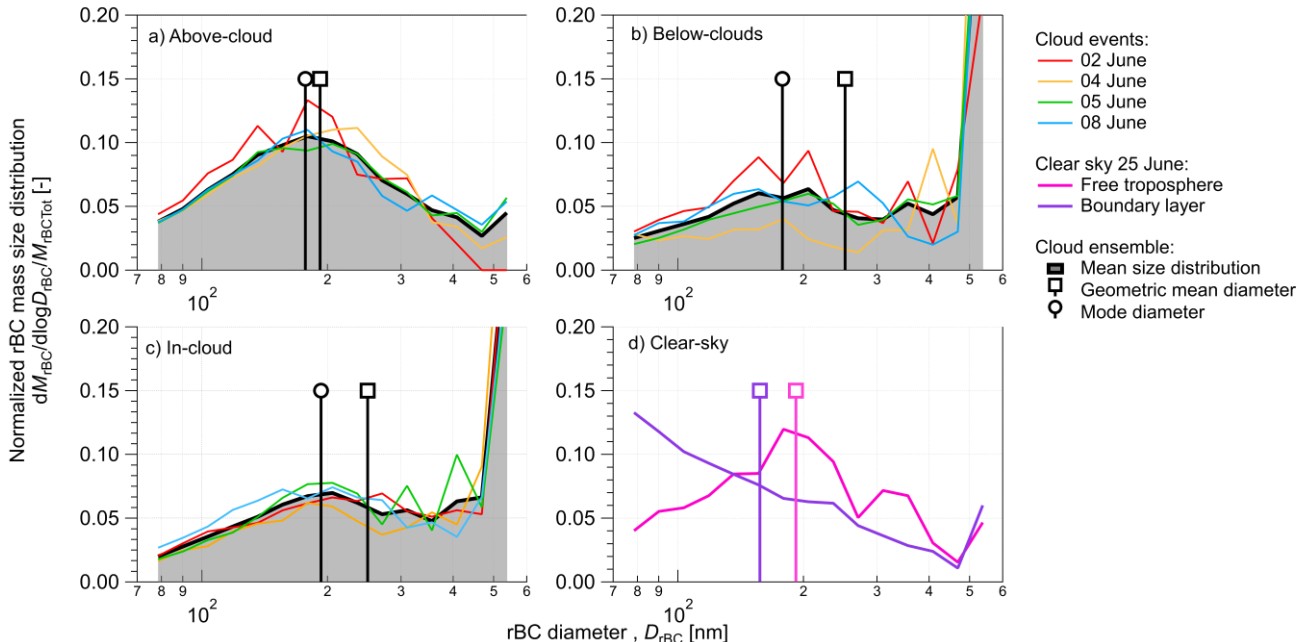

**Figure 4 rBC normalized size distribution observed in: a) above cloud; b) below clouds; c) in cloud; d) clear sky. rBC particles measured in the 75-575 nm diameter range with the SP2. rBC in cloud sampled behind the CVI inlet.**







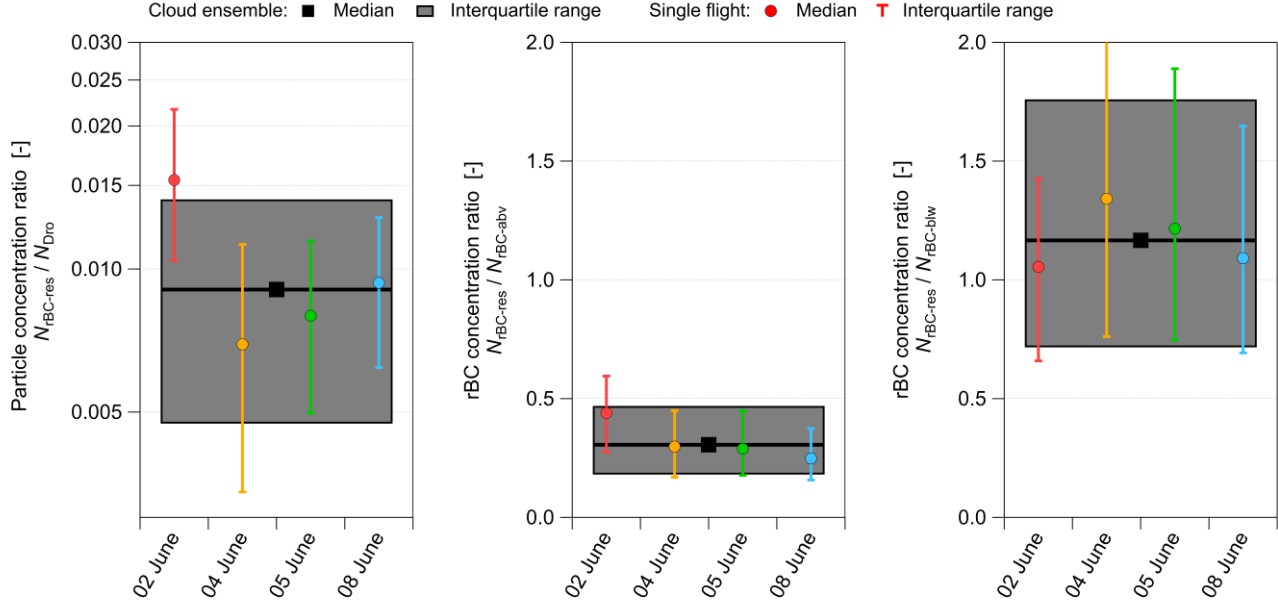

**Figure 5 Box-plot of rBC concentration in cloud residuals. a) fraction of cloud particles containing rBC cores ($CF_{rBC}$) calculated as the ratio of rBC residuals number concentration ($N_{rBC-res}$) over cloud particles number concentration ($N_{Dro}$). b) ratio between the number concentration of cloud residuals (NrBC-res) over the number concentration of rBC particles measured above cloud (NrBC-abv). c) ratio between the number concentration of cloud residuals (NrBC-res) over the number concentration of rBC particles measured below cloud (NrBC-blw). Liquid droplets measured with the SID-3 probe in the 10-45 µm diameter range. rBC residuals sampled behind the CVI inlet or behind the total inlet and measured with the SP2 in the 75-575 nm diameter range.**


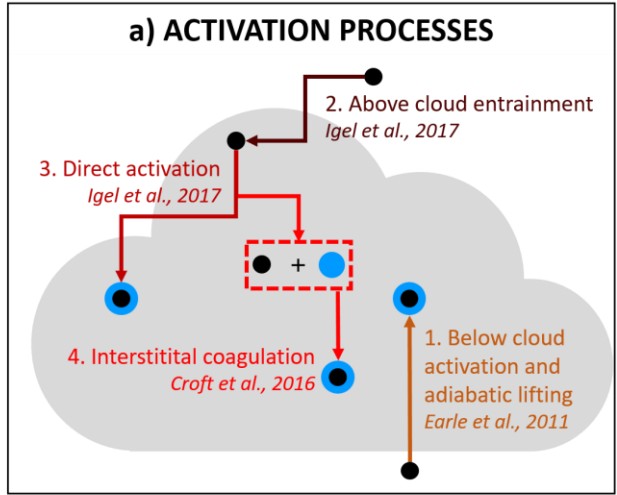

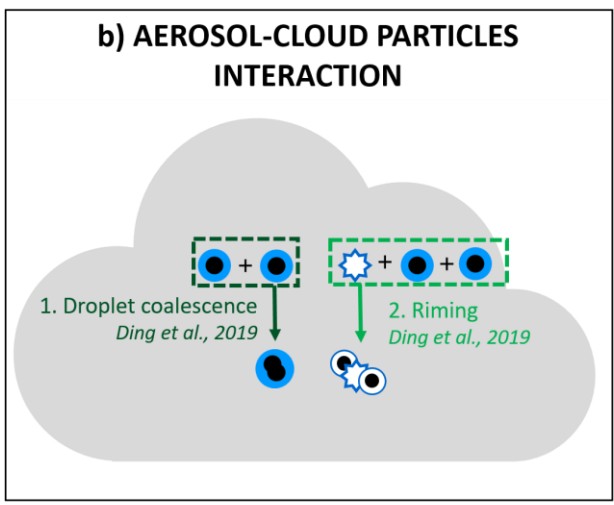

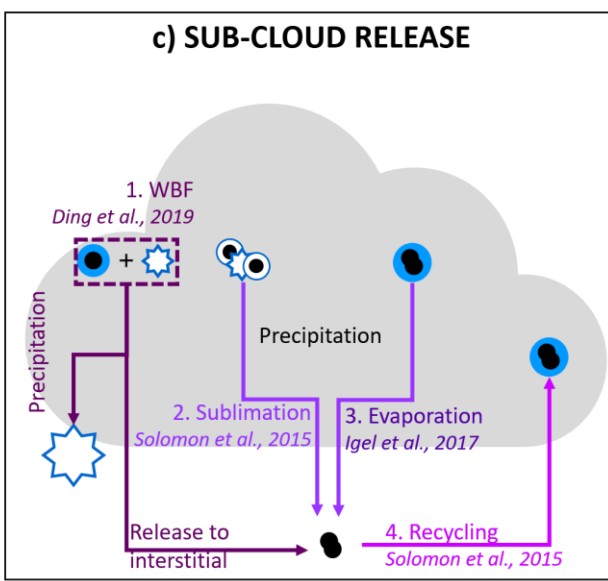

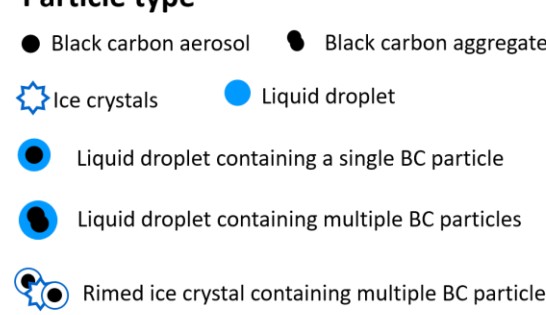

**Figure 6** Potential BC-cloud interaction processes occurring in mixed phase cloud conditions relevant for the Arctic conditions encountered during the ACLOUD campaign. a) BC activation processes in liquid droplets; b) Cloud processes leading to BC metamorphism; c) Processes leading to release of activated BC in the interstitial or aerosol phase.






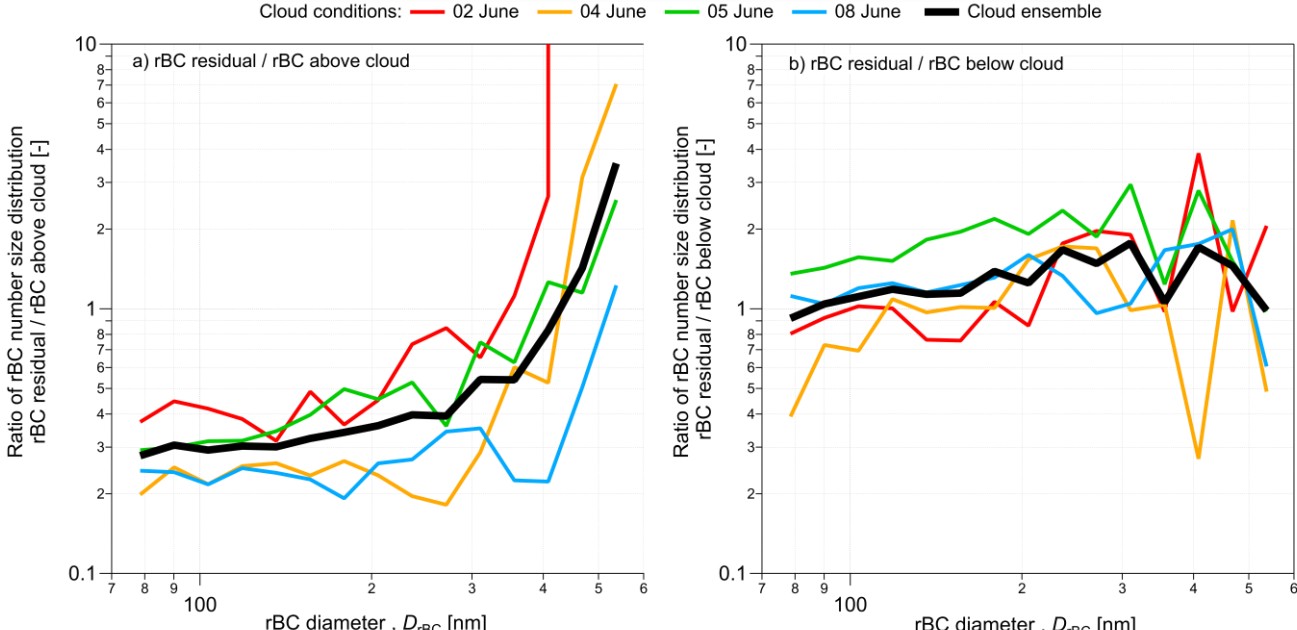


**Figure 7 Ratio of number size distribution of rBC residuals over number size distribution of rBC particles sampled above cloud (a) and below cloud (b). rBC residuals sampled behind the CVI inlet, rBC particles sampled behind the total inlet. All rBC measured with the SP2 in the 75-575 nm diameter range.**





**Figure 8** Vertical variability of cloud particles and rBC residuals in the cloud layer. Liquid droplets: a) number concentration, $N_{Dro}$; b) diameter, $D_{Dro}$; c) liquid water content, LWC. Ice crystals: d) number concentration, $N_{Ice}$; e) diameter, $D_{Ice}$; f) Ice water content, IWC. rBC residuals: g) number concentration, $N_{rBC-res}$; h) diameter, $D_{rBC-res}$. Median and interquartile range calculated for in-cloud equidistant normalized altitude (Zn) steps of 0.2. Liquid droplets measured with the SID-3 probe in the 10-45 µm diameter range. Ice crystals measured with the CIP probe in the 75-1550 µm diameter range. rBC residuals sampled behind the P-CVI inlet and measured with the SP2 in the 75-575 nm diameter range.





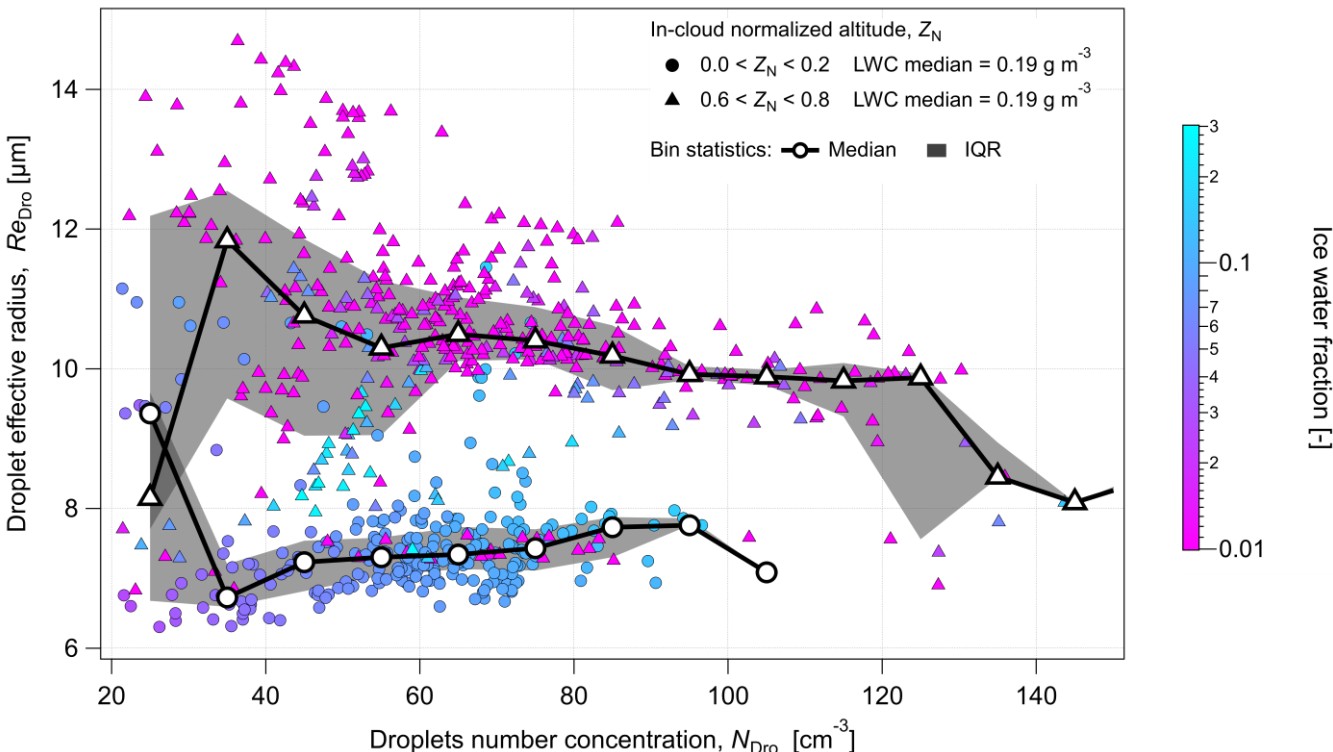

**Figure 9 Relationship between effective radius of liquid droplets and liquid droplets number concentration in different altitudes of the clouds showing the minimum and maximum liquid water content (LWC). Liquid droplets measured with the SID-3 probe in the 10-45 µm diameter range. Median and interquartile range calculated for equidistant liquid droplets number concentration steps of 10 cm⁻³).**