# Peer review of "Airborne investigation of black carbon interaction with low-level, persistent, mixed-phase clouds in the Arctic summer"

_Atmospheric Chemistry and Physics, 2023_

## Referee Comment (RC1)

Manuscript Review

Airborne investigation of black carbon interaction with low-level,

persistent, mixed-phase clouds in the Arctic summer

M Zanatta et al.

**General Comments**

The primary objective of this study is to evaluate rBC properties in cloud hydrometeor residuals and document how these compare under varying synoptic patterns and cloud conditions. A lot of text is created documenting these patterns and conditions, and the authors are quite thorough in the detail that they include describing these patterns and conditions; however, by the end of the manuscript, it becomes quite clear that they are unable to link differences in the observed rBC mass concentrations, size distributions or mixing state to any atmospherically relevant parameter. Put a different way, neither the synoptic patterns or microphysical properties of the clouds are obviously correlated with the rBC properties. Hence, a major portion of the manuscript that describes these patterns and cloud could either be greatly abbreviated or be relegated to the supplement. This includes much of the effort that seems to have gone into describing the SID, SIP, Nevzorov and PHIPS and how they are processed to derive size, shape and concentration since neither LWC, IWC or cloud hydrometeor size distributions are convincingly link to cloud processing of the rBC. In fact, the only features of importance are the regions below, within and above clouds, regardless of their meteorological states. I strongly recommend that the paper be greatly shortened and focus primarily on the possible mechanisms by which the rBC became part of a cloud hydrometeor and only keep those meteorological (T, RH, winds) or cloud (size distributions) parameters in the manuscript that can make the case for or against how these parameters influenced how ambient rBC is processed by the cloud.

**Instrumentation:**

1. Why isn't ice being distinguished in the Small Ice Detector? The high resolution CCD array is specifically designed to allow distinguishing liquid droplets from ice crystals. Why isn't this done? Why make the assumption that the SID only is measuring liquid droplets?
2. Why aren't you looking at both Nevzorov sensing elements to get ice fraction? The Nevzorov consists of a cylindrical LWC hot wire and a heated cup for TWC.
3. Throwing out the first two channels of the CIP is not a generally agreed upon action and doing so is ignoring an important size interval for drizzle.
4. How were ice crystals distinguished in the CIP? It is difficult to determine circularity until you have at least 6 – 8 pixels, i.e. > 200 μm.

5. Why isn't Phips being used to fill in 45-75 fraction?
6. If the cloud concentration/thresholding is from SID, and smallest size is 10 um, you are likely rejecting a lot of cloudy air in you analysis.
7. The SP2 also provides a way to derive the temperature of incandescence to separate light absorbing rBC from other types of material that can absorb at 1024 nm and incandesce. Have the data been properly screened to be sure the incandescing particles are only rBC?
8. The references to Moteki and Laborde for calibration standards and procedures are not the best since Schwarz et al. (2006) was the original publication to which all other subsequent paper refer. Please update.
9. Line 204 "Hence, cloud particle residuals were representative of cloud condensation nuclei and/or ice nucleating particles (Mertes et al., 2005, 2007)." No, this ignores inertial scavenging (see below).

**Data evaluation and presentation:**

1. Show distribution by mass not by derived size. The process of deriving a mass equivalent diameter is highly uncertain because it assumes that all particles with rBC are homogeneous, spherical and constant density. Clearly this is not what in situ samples, captured on filters, reveal. Every where that the rBC distribution by equivalent diameter are shown should be number concentrations or mass concentrations by the measured rBC mass. The same trends and patterns should be seen as are seen with diameter, they will just have more of a physical meaning.
2. The current study looks at ratios of the rBC residual concentrations to the droplet concentrations and rBC residual concentrations to out of cloud rBC concentrations; however, a very important comparison here that is missing should be the comparison of the rBC residual concentrations ratios to the total number of residuals compared with the rBC concentrations out of cloud to the total aerosol concentrations below cloud and above cloud. Baumgardner et al. (2008) used this approach to argue that inertial scavenging was the most likely mechanism for putting rBC in cloud hydrometeor residuals (See below), a mechanism that is almost completely ignored in this manuscript.
3. Where are the representative size distributions from the SID and CIP?
4. These clouds, based on how they are being defined, were between 300 and 400 m in depth, so why did the analysis divide them into 5 layers? I stably stratified clouds like these, is it physically reasonable to think that the properties will have such fine structure?
5. Where are the wind data, vertical and horizontal? Was there vertical shear that would lead to entrainment and mixing? The authors elude to entrainment as a possible mechanism introducing the rBC to the cloud but what drives this entrainment?

Cloud definition:
1.  This study defines a cloud as $N_{Dro} > 10$ cm$^{-3}$ and LWC $> 0.01$ g m$^{-3}$. How was this this threshold arrived at and how sensitive are the results to this definition?
2.  Since $N_{Dro}$ is derived from the SID, and since the lower threshold of the SID is cutoff at 10 µm, in Arctic stratus clouds, this is likely eliminating a lot of cloud data and likely biases measurement depending if they are at the botton half or top half of cloud. This could be part of the reason that you seen differences in the rBC properties at the bottom or top of the cloud.
3.  Where are the size distributions in cloud, compare cloud base to cloud top?

I strongly recommend that the authors read the paper by Baumgardner et al. (2008), who measured rBC in cirrus cystals over the Pacific Ocean, and concluded ***"Comparison of BC properties in the crystal residuals and cloud-free particles show that BC is scavenged by the crystals with an efficiency of as much as 44 ng of carbon per gram of ice water. The average number fraction of BC in crystal residuals was 40% greater than that of the particles in the nearby, cloud-free regions and, on average, the mass equivalent diameter of BC was 10% larger. An average of 24% of the ice crystal residuals contained BC compared to 17% in cloud-free air. The large, in-cloud variability of BC properties prevents a more rigorous proof of our hypothesis that inertial scavenging is a dominant mechanism for removal of BC by clouds. The lack of correlation between the inside and out of cloud concentrations, coupled with the differences between the number fraction and MED in the crystal residuals and the background aerosol, is consistent with the inertial scavenging process and the evidence is sufficiently compelling to warrant further study."***

This study, while sampling a very different type of cloud than the current one, makes the case that all the data points to inertial scavenging as the primary mechanism by which the rBC ends up in cloud hydrometeors. The same arguments can be made in this study. The very limited analysis of the "coating" suggests that these are uncoated or very lighty coated (read "lightly mixed") rBC particles. Laboratory studies, although admittedly sparse and with varying conclusions, generally conclude that fresh, or aged yet lightly coated, rBC are not hygroscopic, i.e. not good CCN; hence, their presence in cloud residual does not imply that they were the nuclei of the droplet. In addition, vertical motion by adiabatic lifting is unlikely to lead to supersaturations large enough to activate these particles, even if slight hygroscopic. The study by Baumgardner et al (2008) claims that that the higher concentrations of larger rBC, seen in their studies and in the present one, is most easily explained by inertial scavenging. If the authors of the current study followed the same methodology as Baumgardner et al., I think that they would like reach a similar conclusion.

To summarize, I think that this manuscript needs to be edited to decrease its size and focus on better explaining the results than are currently being done. I have raised a number questions that need to be addressed before I am ready to recommend this submission for publication.

**References**

Baumgardner, D., R. Subramanian, C. Twohy, J. Stith and G. Kok, **2008**: Scavenging of Black Carbon by Ice Crystals Over the Northern Pacific, *Geophys. Res. Letters*, 35, L22815, doi:10.1029/2008GL035764.

Schwarz, J.P., R. S. Gao, D. W. Fahey, D. S. Thomson, L. A. Watts, J. C. Wilson, J.M., Reeves, D. G. Baumgardner, G. L. Kok, S. Chung, M. Schulz, J. Hendricks, A. Lauer, B. Kärcher, J. G. Slowik, K. H. Rosenlof, T. L. Thompson, A. O. Langford, M. Lowenstein, K. C. Aikin, **2006:** Single-particle measurements of mid latitude black carbon and light-scattering aerosols from the boundary layer to the lower stratosphere, *J. Geophys. Res.* 111, D16207, doi:10.1029/2006JD007076.

---

## Author Response (AR1)

**Acp-2023-30: "Airborne investigation of black carbon interaction with low-level, persistent, mixed-phase clouds in the Arctic summer"**

We would like to thank the referees for their detailed and constructive comments, which helped us to improve our manuscript. While the reviewer's comments are given in black bold, our answers are given below in grey letters. Additionally, we added the **changes made in the revised manuscript in grey bold letters**.

**Answers of the authors to Reviwer#1**

**GENERAL COMMENTS:**
**The primary objective of this study is to evaluate rBC properties in cloud hydrometeor residuals and document how these compare under varying synoptic patterns and cloud conditions. A lot of text is created documenting these patterns and conditions, and the authors are quite thorough in the detail that they include describing these patterns and conditions; however, by the end of the manuscript, it becomes quite clear that they are unable to link differences in the observed rBC mass concentrations, size distributions or mixing state to any atmospherically relevant parameter. Put a**

20 **different way, neither the synoptic patterns or microphysical properties of the clouds are obviously correlated with the rBC properties.**
**Hence, a major portion of the manuscript that describes these patterns and cloud could either be greatly abbreviated or be relegated to the supplement. This includes much of the effort that seems to have gone into describing the SID, SIP, Nevzorov and PHIPS and how they are processed to derive size, shape and concentration since neither LWC, IWC or cloud hydrometeor size distributions are convincingly link to cloud processing of the rBC. In fact, the only features of importance are the regions below, within and above clouds, regardless of their meteorological states. I strongly recommend that the paper be greatly shortened and focus primarily on the possible mechanisms by which the rBC became part of a cloud hydrometeor and only keep those meteorological (T, RH,**

30 **winds) or cloud (size distributions) parameters in the manuscript that can make the case for or against how these parameters influenced how ambient rBC is processed by the cloud.**
The manuscript was modified according to the reviewer's suggestion:

- The cloud-microphysics data were reduced to a minimum. The liquid water content derived from the SID-3 and the ice water content derived from the CIP are now used to generally describe the dominant phase of the cloud. As a consequence, the technical description provided in Section 2.2.2 was also shortened.
- Inertial scavenging was investigated with the analytical approach of Baumgardner et al. (2008) and discussed in Section 3.4.1. Most likely due to the different nature of clouds observed in this study compared to Baumgardner et al. (2008), we concluded that rBC particles and the bulk aerosol were

40 activated following a similar activation process.
- The mostly speculative Section 4 was removed, while the discussion on the in-cloud vertical variability now focusses, mostly, on cloud-top and cloud-bottom sections.

With these major changes we gave more space to the discussion of : 1) presence and properties of cloud-active BC particles; 2) scavenging mechanisms; 3) BC residuals vertical distribution.

**INSTRUMENTATION:**
**1. Why isn't ice being distinguished in the Small Ice Detector? The high resolution CCD array is specifically designed to allow distinguishing liquid droplets from ice crystals. Why isn't this done? Why make the assumption that the SID only is measuring liquid droplets?**

50    In the updated ACLOUD dataset, the two dimensional scattering signal of the SID-3 probe was analysed for particles sphericity (Vochezer et al., 2016) and crystal complexity (Schnaiter et al., 2016). More specific details can be found in Järvinen et al. (2023). Now, $N_{Dro}$ refers only to liquid droplets quantified in the 5-45 µm diameter range (See more details on the size segregation of liquid droplets in the answer to comment number 6). Due to the low number concentration of ice crystals detected by the SID-3 (Figure A), the statistics of $N_{Dro}$ did not change significantly in the latest version of the manuscript. More details on the SID-3 data treatment and definition of cloud boundaries are given in the answers to comments number 6 and number 16. In the description of the SID-3 in Section 2.2.2 the comparison with the Nevzorow was removed and the text modified as:

**… "The Small Ice Detector Mark 3 (SID-3; Hirst et al., 2001; Vochezer et al., 2016) allows deriving the**
60    **cloud particle number size distribution in the 5-45 µm diameter range (https://doi.org/10.1594/PANGAEA.900261; Schnaiter and Järvinen, 2019b). Liquid droplets and ice crystals were distinguished based on analysing the SID-3 two-dimensional (2-D) scattering patterns for the particle sphericity as described in Vochezer et al., 2016. A more detailed description of processing the SID-3 data for ACLOUD including correction of coincidence artefacts can be found in Järvinen et al. (2023). In the present work, the number concentration ($N_{Dro}$) and diameter ($D_{Dro}$) of liquid droplets was calculated in the 10 - 45 µm diameter range to match the low size cut-off of the CVI inlet (10 µm). The liquid water content (LWC) was calculated from the size distribution assuming spherical particles and a particle density of 1 g cm$^{-3}$. Due to the scarcity of ice in the 10 - 45 µm diameter range, ice crystals measured by the SID-3 are not discussed in the present work." …**

[Figure]

70    *Figure A Comparison between droplet number concentration ($N_{Dro}$) and total cloud particle concentration ($N_{Dro+Ice}$; droplet and ice) for the cloud events presented in the manuscript (02-08 June 2017).*

**2. Why aren't you looking at both Nevzorov sensing elements to get ice fraction? The Nevzorov consists of a cylindrical LWC hot wire and a heated cup for TWC.**
Unfortunately, due to the uncertainties in total water content and liquid water content measured with the Nevzorov probe, the difference method could not be used to calculate ice water content, which had the same order of magnitude than the uncertainties (Ehrlich et al., 2019). In view of this aspect
80    and considering that Nevzorov data were not used in the results section, the Nevzorov probe is not described any longer in the method section, while the comparison with the SID-3 LWC was removed.

**3. Throwing out the first two channels of the CIP is not a generally agreed upon action and doing so is ignoring an important size interval for drizzle.**
The particle concentrations for sizes below 100 μm are highly uncertain due to the presence of larger particles which are out of focus. Also, out-of-focus image corrections are more uncertain the lower the number of pixels. Due to this uncertainty in the first classes, we have decided to remove the first two channels of the CIP instrument.

90 **4. How were ice crystals distinguished in the CIP? It is difficult to determine circularity until you have at least 6 – 8 pixels, i.e. > 200 μm.**
The validation criterium of circularity was based on area rather than diameter. We followed Crosier et al. (2011) using Particle Size Distribution (maximum diameter – all in focus) and a circularity > 1.25 only for area larger than 16 pixels. Obviously, that does not include particles with sizes smaller than 125 μm.

**5. Why isn't Phips being used to fill in 45-75 fraction?**
Following the comments of both reviewers, the discussion about cloud microphysics was simplified to focus more on rBC activation processes. Considering that the larger ice crystals detected by the CIP
100 (75-1550 μm) would have a major contribution to ice water content compared to smaller ice crystals (D<75μm), the CIP data provided sufficient information to generally investigate the cloud-phase (ice water content) in the new Section 3.6.2. In order to shorten the manuscript PHIPS data were removed entirely from the manuscript since they only contained non-essential information.

**6. If the cloud concentration/thresholding is from SID, and smallest size is 10 um, you are likely rejecting a lot of cloudy air in you analysis.**
The treatment of SID-3 data was significantly modified.
- Phase discrimination was applied. Now, $N_{Dro}$ refers only to liquid droplets.
- For the identification of cloud boundaries and the characterization of cloud phase, the size-cut of
110 10 mm was removed. Now, $N_{Dro}$ and LWC were derived from the SID-3 probe in the 5-45 μm range.
- For residuals measurements, the number concentration of liquid droplets was calculated between 10 μm and 45 μm to match the low size-cut of the CVI (as in the previous version of the manuscript). To avoid confusion, the previous nomenclature "$N_{Dro}$" was modified into "$N_{Dro10}$".

**7. The SP2 also provides a way to derive the temperature of incandescence to separate light absorbing rBC from other types of material that can absorb at 1024 nm and incandesce. Have the data been properly screened to be sure the incandescing particles are only rBC?**
Since both reviewers raised this point, we address it in full details as it follows.
- COLOUR-RATIO: Different black carbon or soot types are characterized by a different boiling
120 temperature, which can be derived from the ratio of the intensity of the incandescence signal detected by the broad-band detector (wavelength detection range of 300-800 nm) over the narrow-band detector (wavelength detection range of 630-800 nm). This ratio is commonly called "colour-ratio". Schwarz et al. (2006) showed how different soot types are characterized by different boiling-point temperatures. Dahlkötter (2014) showed an increase of the colour ratio from background conditions (values around 1.6-1.0) to biomass burning conditions (values around 2.0-1.5). So, the SP2 can be used to discriminate between different types of rBC.
- MINERAL-DUST INTERFERENCE: As shown by Schwarz et al. (2006), the SP2 is also sensitive to metal containing particles. This feature is nowadays exploited to derive the concentration of iron oxides with a modified SP2, based on non-overlapping detection range of two incandescence
130 detectors (blue wavelength detection range of 300-550 nm; red wavelength detection range of 580-710 nm; Yoshida et al., 2016). According to the latter, hematite and magnetite show a distinct colour ratio (~1.5) compared to rBC standard material (fullerene; 2.4). Considering the different

wavelength detection range, the values presented by Yoshida et al. (2016) cannot be directly compared with ACLOUD or Dahlkötter (2014) measurements.

- COLOUR-RATIO DURING ACLOUD: The colour-ratio was calculated for the ACLOUD campaign: above-cloud median=1.49 (IQR=1.38-1.62), inside-cloud median=1.45 (IQR=1.31-1.66) and below-cloud median = 1.44 (IQR=1.30-1.66) cloud. The colour-ratio analysis was also performed as function of mas equivalent diameter (Figure B). Similar to Dahlkötter (2014), we observed a decrease of the colour-ratio with increasing rBC mass equivalent diameter. However, we did not observe a clear difference between the above-cloud, inside-cloud and below-cloud measurements, which allowed for some clear screening of incandescing but yet non-rBC particles such as iron oxides as in Yoshida et al. (2016, 2020).

- SCREENING OF INVALID PARTICLES: Following the analysis performed in Yoshida et al. (2016), we tried to identify and remove those large particles (DrBC>300nm) showing prolonged incandescence signals, which might be related to mineral dust (Figure C). A signal was considered "valid" only if its duration was shorter than 30 us and the rise-time was shorter than decay time. Overall, a minor fraction of the total particles larger than 300 nm were considered to be invalid. The invalid particles did not show a peculiar colour-ratio which could be related to mineral dust and had a negligible effect on the size distributions presented in the manuscript. Nonetheless, all the invalid particles were removed.

- TEXT CHANGES: A short statement was added to Section 2.2.3 and reads:

… **"Previous studies showed that the SP2 is sensitive to metal containing particles such as hematite and magnetite, which might lead to an overestimation of rBC particle concentration (Schwarz et al., 2006; Yoshida et al., 2016). These particles are characterized by lower boiling point and colour-ratio (the ratio of thermal emission in the blue and red spectrum) but also by slow heating-rate in the laser beam of the SP2. During ACLOUD, particles associated with slow rise-time of the incandesce signal were removed. The colour-ratio analysis did not show any clear evidence of the presence of non-rBC yet incandescing particles."** …

[Figure]

*Figure B Diameter dependency of the color-ratio calculated for all incandesce signal observed above-cloud, inside-cloud and below-cloud with the SP2.*

[Figure]

*Figure C Incandescence signal of "valid" and "invalid" particle.*

**8. The references to Moteki and Laborde for calibration standards and procedures are not the best since Schwarz et al. (2006) was the original publication to which all other subsequent paper refers. Please update.**

The reference was updated.

**9. Line 204 "Hence, cloud particle residuals were representative of cloud condensation nuclei and/or ice nucleating particles (Mertes et al., 2005, 2007)." No, this ignores inertial scavenging (see below).**

We implemented the data analysis suggested by the reviewer and concluded that inertial scavenging was not relevant for the ACLOUD cases. None the less, inertial scavenging and its importance in the Arctic is now better described in the introduction and in the result section. Please find more details in the answers to comment number 21.

**DATA EVALUATION AND PRESENTATION:**

**10. Show distribution by mass not by derived size. The process of deriving a mass equivalent diameter is highly uncertain because it assumes that all particles with rBC are homogeneous, spherical and constant density. Clearly this is not what in situ samples, captured on filters, reveal. Everywhere that the rBC distribution by equivalent diameter are shown should be number concentrations or mass concentrations by the measured rBC mass. The same trends and patterns should be seen as are seen with diameter, they will just have more of a physical meaning.**

In the SP2-specific literature, there is an overall cohesion on the use of a fixed bulk density of 1800 kg/m$^3$ presented by Moteki et al. (2010) to convert mass into mass equivalent diameter. Bearing in mind the general consistency in SP2 literature, changing the size distribution into mass distribution would not modify the results of our study nor improve its understanding; but, on the contrary, limit the comparability with previous and future studies. Hence, the size distributions presented in the manuscript were not modified.

**11. The current study looks at ratios of the rBC residual concentrations to the droplet concentrations and rBC residual concentrations to out of cloud rBC concentrations; however, a very important comparison here that is missing should be the comparison of the rBC residual concentrations ratios to the total number of residuals compared with the rBC concentrations out of cloud to the total aerosol concentrations below cloud and above cloud. Baumgardner et al. (2008) used this approach to argue that inertial scavenging was the most likely mechanism for putting rBC in cloud hydrometeor residuals (See below), a mechanism that is almost completely ignored in this manuscript.**

See answer to comment number 21.

**12. Where are the representative size distributions from the SID and CIP?**
Considering that the discussion of SID and CIP data was drastically reduced, the size distribution at cloud-top and cloud-bottom are briefly discussed in Section 3.6.3 and shown in Figure S5.

**13. These clouds, based on how they are being defined, were between 300 and 400 m in depth, so why did the analysis divide them into 5 layers? I stably stratified clouds like these, is it physically reasonable to think that the properties will have such fine structure?**
The vertical profiles presented in former Figure 8 are now presented into 4 vertical sections. Former Figure 8 was also greatly reduced, while more emphasis is now given to the differences between cloud top and cloud bottom in Section3.6.3.

**14. Where are the wind data, vertical and horizontal? Was there vertical shear that would lead to entrainment and mixing? The authors elude to entrainment as a possible mechanism introducing the rBC to the cloud but what drives this entrainment?**
We agree with the reviewer that wind velocity would help to understand the activation mechanism. As described in Ehrlich et al. (2019), the vertical wind can only be analysed as the deviation from the average for flight sections of at least several kilometres long, ideally for straight and levelled flight sections. The saw-tooth profile often adopted during the 4 flights presented here did not provide enough horizontal flight-time for a statistically robust analysis. Unfortunately, due to this technical limitation, wind data could not be included into the analysis.

**CLOUD DEFINITION**
**15. This study defines a cloud as NDro > 10 cm-3 and LWC > 0.01 g m-3. How was this this threshold arrived at and how sensitive are the results to this definition?**
See answer to comment number 16.

**16. Since NDro is derived from the SID, and since the lower threshold of the SID is cutoff at 10 μm, in Arctic stratus clouds, this is likely eliminating a lot of cloud data and likely biases measurement depending if they are at the botton half or top half of cloud. This could be part of the reason that you see differences in the rBC properties at the bottom or top of the cloud.**
As a follow up of reviewer's comment number 6, the screening of SID-3 data was modified by removing the 10 μm cut-off. Now, the full detection range of the SID-3 (5-45 μm) was used to quantify $N_{Dro}$ and LWC, and to identify the cloud boundaries. In the updated version of the manuscript, the cloud LWC-threshold was kept to 0.01 g m$^{-3}$ (Järvinen et al., 2023), while the cloud $N_{Dro}$-threshold was set to 1 cm$^{-3}$. These new cloud-thresholds did not impact, significantly the time of valid in-cloud measurements (difference of less than 2 minutes on the cloud ensemble) and the overall rBC variability observed during ACLOUD. Results in the figures and in the text were updated accordingly. Note that the threshold for outside-cloud measurements was not modified.

**17. Where are the size distributions in cloud, compare cloud base to cloud top?**
Former section 3.5 was modified following the comments of both reviewers. It now includes a short description of potential temperature, liquid water content, ice water content and rBC-residuals number concentration. In the second part of the section, we compare in more details the properties of rBC-residuals and cloud particles observed at cloud-top and cloud-bottom. Size distribution of rBC-residuals are shown in the manuscript, while size distribution of liquid droplets and ice crystals are shown in the supplementary.

**18. I strongly recommend that the authors read the paper by Baumgardner et al. (2008), who measured rBC in cirrus crystals over the Pacific Ocean, and concluded "Comparison of BC properties**

**in the crystal residuals and cloud-free particles show that BC is scavenged by the crystals with an efficiency of as much as 44 ng of carbon per gram of ice water. The average number fraction of BC in crystal residuals was 40% greater than that of the particles in the nearby, cloud-free regions and, on average, the mass equivalent diameter of BC was 10% larger. An average of 24% of the ice crystal residuals contained BC compared to 17% in cloud-free air. The large, in-cloud variability of BC properties prevents a more rigorous proof of our hypothesis that inertial scavenging is a dominant mechanism for removal of BC by clouds. The lack of correlation between the inside and out of cloud concentrations, coupled with the differences between the number fraction and MED in the crystal**

260 **residuals and the background aerosol, is consistent with the inertial scavenging process and the evidence is sufficiently compelling to warrant further study." This study, while sampling a very different type of cloud than the current one, makes the case that all the data points to inertial scavenging as the primary mechanism by which the rBC ends up in cloud hydrometeors. The same arguments can be made in this study.**

We carefully read Baumgardner et al. (2008), and applied their analysis to investigate inertial scavenging. The reviewer can find more details in the answer to comment number 21.

**19. The very limited analysis of the "coating" suggests that these are uncoated or very lighty coated (read "lightly mixed") rBC particles.**

270 The coating thickness and shell-to-core diameter ratio fell in a similar range of previous Arctic studies (Raatikainen et al., 2015; Zanatta et al., 2018; Schulz et al., 2019; Yoshida et al., 2020; Ohata et al., 2021). To the other end, freshly emitted rBC-containing particles are associated with shell-to-core around 1.1 (Laborde et al., 2013; Yoshida et al., 2020). As shown in Figure D (now included in the supplementary), coating thickness up to140 nm were observed inside and below cloud. By comparing ACLOUD data with the results of the studies presented above, rBC particles observed during ACLOUD could be classified as "medium-coated" to "thickly-coated" but not as "uncoated" or "very lightly coated" as stated by the reviewer.

[Figure]

*Figure D Coating thickness distribution calculated for rBC particles with a mass equivalent diameter in*
280 *the 200-250 nm range. Coating thickness data available only for the flight occurred on 02 June 2017.*
*Added to the supplementary material.*

**20. Laboratory studies, although admittedly sparse and with varying conclusions, generally conclude that fresh, or aged yet lightly coated, rBC are not hygroscopic, i.e. not good CCN; hence, their presence in cloud residual does not imply that they were the nuclei of the droplet. In addition,**

**vertical motion by adiabatic lifting is unlikely to lead to supersaturations large enough to activate these particles, even if slight hygroscopic.**

Works performed by Schwarz et al. (2015) and Ohata et al. (2016) clearly show, as the reviewer suggest, that uncoated rBC particles (shell-to-core ratio <1.1) are not hygroscopic. Formation of coatings may, however, increase the hygroscopicity of rBC particles. In fact, 70% rBC particles with coating thickness of 40 nm may be activated in liquid droplets already at a peak SS of 0.15-0.20% (Motos et al., 2019a), while Dalirian et al. (2018), demonstrated that formation of relatively thin organic coatings (shell-to-core ratio between 1.15 and 1.30) drastically decreases the critical supersaturation. Hence coating thickness above 30 nm are sufficient to increase the hygroscopicity of rBC particles and promote nucleation scavenging. The rBC sampled during ACLOUD showed coating thickness up to 140 nm with median between 30-45 nm, enough to be classified as aged and hygroscopic. Furthermore, the number fraction analysis, suggested by the reviewer, indicated that the bulk aerosol and rBC particles were activated following the same scavenging mechanism (see below for more details).

- To better contextualize the importance of coating thickness, the introduction was modified as:

  … **"The ability of BC particles to promote droplet formation (nucleation scavenging) is one of the most complex parametrisations in global model schemes. If fresh BC particles are not hygroscopic, aged BC particles shows an enhanced hygroscopicity (Schwarz et al., 2015; Ohata et al., 2016). In fact, the nucleation ability of BC depends on fundamental particle properties such as diameter and mixing state, which change due to intra-coagulation, and to condensation and coagulation with other atmospheric species. Hygroscopicity increases with particles size (Motos et al., 2019a) and the formation of inorganic and organic coatings (Dalirian et al., 2018; Motos et al., 2019b)."** …

- Section 3.5.1 was fully updated to better present the coating distribution and its potential impact on the hygroscopicity during ACLOUD:

  … **"First it must be noted that the analysis of the coating thickness includes only rBC particles in the 200-250 nm range of mass equivalent diameter. rBC particles in this diameter range were ubiquitously found above-cloud, inside-cloud and below-cloud (Figure 3). The distribution of coating-thickness is presented in Figure S2 in the supplementary material. The thinnest coatings were observed above clouds, where the coating thickness median was 30 nm (IQR = 23 - 48 nm) and median shell-to-core ratio was 1.51 (IQR = 1.38 - 1.8). The thickest coatings were observed below clouds, where the median coating thickness was 43 nm (IQR = 25 - 58 nm), and median shell-to-core diameter ratio was 1.67 (IQR = 1.43 - 1.98). The rBC cloud residuals showed medium coating thickness (median = 38 nm, IQR = 25 - 59 nm) and shell-to-core ratio (median = 1.58, IQR = 1.39 − 1.92 nm) respect to above-cloud and coatings to below-cloud. The coating thickness values presented here are similar to previous Arctic ground (Raatikainen et al., 2015; Zanatta et al., 2018) and airborne (Kodros et al., 2018; Ohata et al., 2021) observation, and are substantially higher than urban observations (Laborde et al., 2013; Yoshida et al., 2020). Even though thicker coatings can be found in aged continental air masses, the presence of 30-40 nm thick coatings is sufficient to significantly increase the hygroscopicity of otherwise hydrophobic uncoated BC particles in laboratory experiments (Dalirian et al., 2018) and filed observations (Motos et al., 2019a). Keeping in mind the low counting statistics of the coating analysis, we can conclude that rBC particles sampled during ACLOUD were presentative of aged and hygroscopic rBC particles which could be efficiently activated via nucleation scavenging. However, the reduced temporal coverage and the uncertainty of coating thickness quantification (17%; Laborde et al., 2012) did not allow identifying a significant change in the degree of internal mixing between rBC residuals and rBC particles sampled outside cloud."** …

**21. The study by Baumgardner et al (2008) claims that that the higher concentrations of larger rBC, seen in their studies and in the present one, is most easily explained by inertial scavenging. If the**

**authors of the current study followed the same methodology as Baumgardner et al., I think that they would like reach a similar conclusion.**

Interstitial or inertial scavenging was clearly overlooked in our work. Hence, following the reviewer's suggestion we calculated the rBC number fraction ($F_{rBC}$) by combining the rBC number concentration (SP2 measurement) and total aerosol number concentration (UHSAS measurements). The number fraction of rBC particles was extremely low below-cloud (mean=1.4%), inside-cloud (mean=1.1%) and above-cloud (mean=2.5%). These values being 1 order of magnitude lower than the values reported in Baumgardner et al. (2008) (mean in-cloud=24%, mean outside-cloud=17%). During ACLOUD, the fraction of rBC particles was not enriched in clod residuals, as in in Baumgardner et al. (2008), but remained rather constant. By remaining substantially constant between outside and below-cloud, $F_{rBC}$ indicates that rBC is activated in the same ratio as other particles not containing rBC cores, thus sharing a similar activation pattern and, perhaps, hygroscopicity.

- To better present interstitial scavenging, a new paragraph was added to the introduction section:

**… "Moreover, other in-cloud processes might compete with nucleation scavenging. In fact, interstitial BC particles (BC particles present in the cloud volume but not activated into cloud particles) can be efficiently captured by pre-existing cloud particles via interstitial scavenging (Baumgardner et al., 2008). Despite being often ignored, the number concentration of Arctic aerosol is highly sensitive to interstitial scavenging occurring during long-range transport (Croft et al., 2016)." …**

- The analysis of rBC number fraction is now presented in the new Section 3.4.2:

**… "Other activation mechanisms might occur and contribute to the $N_{rBC-res}/N_{rBC-blw}$ values above unity. First, interstitial aerosol particles may be scavenged via impaction with existing droplets (Croft et al., 2016), potentially enriching the number concentration of rBC-residuals (Baumgardner et al., 2008). We thus compared the fraction of rBC particles measured outside clouds ($F_{rBC}= N_{rBC}/N_{AP}$) and inside clouds ($F_{rBC-res}=N_{rBC-res}/N_{AP-res}$). An increase of $F_{rBC-res}$ compared to outside cloud, might indicate the preponderant activation of rBC particles via interstitial scavenging (Baumgardner et al., 2008). During the ACLOUD cases, we found slightly smaller $F_{rBC-res}$ (1.0%) than $F_{rBC}$ above-cloud (2.3%) and below-cloud (1.2%). Similar $F_{rBC-res}$ and $F_{rBC}$ below-cloud suggested that rBC was activated via the same pathway of the bulk aerosol, that rBC and other aerosol particles shared similar hygroscopicity, and that rBC particles were not preferentially activated by interstitial scavenging. "…**

**Answers of the authors to Reviwer#2**

**SUMMARY:**
**… I would like to see the authors to be more constrained with the ambition and focus on what can be said about rBC in Arctic could drops given the present data. I encourage the authors to leave out the sometimes-lengthy speculative parts of the manuscript; i.e. lengthy speculations about processes that cannot be addressed with these data. The manuscript could use more cohesion. It feels like the introduction is lists studies where particular phenomena have been studied and those studies have come to the listed conclusions. I would expect the authors to connect these studies in an easier way so that it is easy to follow and would subsequently open up to the reader why the authors chose to cite those articles; and why the chosen articles are relevant for the present study. Throughout the manuscript, references to previous work is done in a fashion that the authors expect the reader to know them by hearth, and sometimes mid-sentence without an obvious connection to the present work. I would encourage the authors to consider which references they want to refer to, and only refer to previous work when they are obviously relevant. That would draw more attention to your work and why it is important instead of stating that someone already did something similar in another part of the world.**

The manuscript was reworked considering the reviewer's comments, putting particular care in harmonizing the scientific message and removing the speculative parts. The objectives are now clearly listed in the introduction and include:

1. Quantify the presence and properties of cloud-active BC particles
2. Identify the main scavenging mechanism
3. Understand the impact of cloud processing on BC vertical distribution

Former Section 4 was fully removed together with former Figure 6 and Figure 9. The cloud microphysics data, formerly very detailed, are now reduced to a minimum leaving more space to the discussion and interpretation of black carbon data. The narrower focus on BC also allowed reducing the number of citations and address them in more detail.

**GENERAL COMMENTS:**
**The nomenclature is not always consistent in the manuscript which sometimes makes the manuscript unnecessarily hard to read. Please make sure that the same terminology is used throughout the manuscript. E.g. use either CCN or cloud drop residual, not both. Use wet scavenging or cloud scavenging, or make a clear distinction what you mean if you refer to below-cloud or in-cloud scavenging or something else.**

The nomenclature was thoughtfully controlled and harmonized throughout the manuscript. Table 1 was modified accordingly.

**Please check the use of "this", "those" and "these" in the text. It can be hard to follow to what the word is referring to. Please, repeat the statement or the results (in brief) instead of referring loosely to what was said before. That would make for easier reading. Examples P11L343 "Under this cloud regime…", P14L442 "Under this very complex..."**

The issue was addresses.

**For the above stated reasons, I suggest that the manuscript bee distilled down to comprise results and discussion on Figures 1-5, 7.**

The structure of the manuscript was modified substantially by reducing the overall length, by rearranging the structure to discuss scavenging mechanisms (Section 3.4 and subsectons) and vertical variability (Section 3.6 and subsections), and by removing the lengthy discussion on cloud processing.

**The conclusions section summarizes the present study well and is a good reference for how the work should be shortened.**

This comment was particularly helpful, thanks.

**SPECIFIC COMMENTS**

**Section 2: It is never mentioned how one single SP2 measures both "rBC cloud residuals" and rBC particles outside of clouds behind the CVI. Two inlets are mentioned but only one inlet is described in detail.**

The SP2 and UHSAS were sampling through a line "shared" by both the CVI and total inlet. In normal operation outside clouds, both instruments were sampling exclusively though the total inlet. During in-cloud transects, the SP2 and UHSAS were manually connected to the CVI inlet by a valve switch. A short description was added at the end of Section 2.3 and now reads:

…**"SP2 and UHSAS were operated in parallel and shared a sampling line which was alternatively connected to the total inlet or the CVI inlet. Outside cloud ($N_{Dro}$ = 0 cm$^{-3}$ and LWC = 0 g m$^{-3}$), SP2 and UHSAS measurements were performed at the aerosol inlet. Inside cloud ($N_{Dro}$ ≥ 1 cm$^{-3}$ and LWC ≥ 0.01 g m$^{-3}$), the SP2 and UHSAS were sampling throughout the CVI inlet line."**…

**Section 3.5 and Figure 8: Different low-level clouds have been scaled to the same thickness (rage 0 – 1 where 0 is the cloud bottom and 1 is the cloud top). These data comprise low exclusively level clouds, right? I think it is misleading to talk about cloud layers inside the "ensemble" cloud. A cloud layer is a set of clouds at some altitude and makes a cloud layer when it is distinguishable from another cloud layer with cloud free skies in between. To talk about a cloud layer inside a cloud does not make sense to me. How about talking about e.g. the lowermost 20% of the "ensemble cloud" or uppermost 60% etc? Or something better that you come up with**

We agree with the reviewer. Note that following reviwer#1's comment, the vertical profiles were arranged in 4 and not 5 vertical sections. The word "layer" was replaced with "quartile" or "vertical section". In Section 3.6.3, the top quartile and bottom quartile are more simply referred as "cloud-top" and cloud-bottom", respectively.

**I would suggest removing the Discussion section (section 4) and compress the results contents to one short paragraph and leave out the speculation.**

This comment was shared by reviewer#1. Former Section 4, Figure 6 and Figure 9 were removed from the manuscript. Some elements of Section 4 were briefly elaborated in the introduction and Section 3.4.2 and Section 3.6.3.

**When talking about the size of the rBC particles in the text, make sure you state if you are talking about the mode of the size distribution, mass or geometric mean diameter or something else. It is not always clear what size is meant.**

Following the reviewer's suggestion, the very generic "rBC diameter" term was replaced with the more specific "rBC mass-equivalent diameter" term ($D_{rBC}$). The change includes all the text and Figure 2,3,5,7. We also introduced two new acronyms: $D_{rBC-mod}$ refers to the mode diameter of the rBC size distribution; $D_{rBC-geo}$ refers to the geometric mean of the rBC size distribution. Table 1 was modified accordingly. A short statement was added in section 2.2.3 and reads:

… **"The geometric mean and modal diameter of the mass size distribution will be abbreviated as $D_{rBC-Geo}$ and $D_{rBC-Mod}$, respectively."** …

**P1L25 "might suggest" is not that intriguing. Rephrase to raise interest.**

The abstract was considerable reworked, and the statement in question modified as:

… **"The vertical evolution of rBC properties from inside-cloud and below-cloud indicated an efficient aerosol exchange at cloud-bottom, which might include activation, cloud processing, and sub-cloud release of processed rBC agglomerates."** …

**P1L26 It would be interesting to give the reader a hint of the evidence for this kind of processing already here.**

In the abstract, we now summarized the evidences in support to our conclusion.

**P2L46-47 rBC after carbon dioxide and methane makes rBC 3[rd]. Please rephrase or clarify.**

Corrected:

… **"making BC the third atmospheric Arctic warmer only after the trace gases carbon dioxide and methane"** …

**P2L49-50: "Precipitation occurring during long-range transport influences the seasonal cycle of BC" I suggest saying something about the sources of rBC in the Arctic somewhere close to this text. Otherwise it is not that clear why deposition during long range transport has such an impact on rBC concentrations in the Artic.**

The sentence was modified as:

… **"The seasonality of BC concentration at the Arctic surface is characterized by a maximum in the early spring and a minimum in summer (Quinn et al., 2015). A similar seasonality was also recently reported on the vertical scale (Jurányi et al., 2023). Due to the scarcity of BC sources within the Arctic, most of BC mass reaches the Arctic via long range transport (Xu et al., 2017). Hence, the seasonal cycle is mostly controlled by the circulation of air masses between the Arctic and southern latitudes (Bozem et al., 2019), and precipitation intensity during long-range transport (Croft et al., 2016)."** …

**P2L54-55 "...might easily be the limiting process..." is too vague.**

The introduction was substantially reworked. This specific section now reads:

… **"Overall, cloud scavenging is responsible of 90% of BC mass deposition in the Arctic (Dou and Xiao, 2016), with the highest precipitation rate in summer contributing to a decline of BC burden from late spring to autumn (Garrett et al., 2011; Mori et al., 2020)."**…

**P2L62-P3L65 Unclear and too long sentence. Too many references to too many studies makes for a difficult sentence to follow. "No observations... aforementioned studies ... indicating ... size distribution of (cloud drop size? Aerosol size?) ... degree of internal mixing..."**

See answer to following comment.

**P3L65 A bit unclear sentence and could be made easier to follow. As it is now, the reader must know that CCN and hygroscopicity are connected, that fresh soot is small in size.**

The introduction was substantially reworked. This specific section now reads:

… **"The ability of BC particles to promote droplet formation (hygroscopicity) is one of the most complex parametrisations in global model schemes. In fact, the cloud nucleation ability of BC depends on fundamental particle properties such as diameter and mixing state, which change during atmospheric ageing due to condensation and coagulation processes. If fresh BC particles are not hygroscopic, aged BC particles show an increase of hygroscopicity (Schwarz et al., 2015; Ohata et al., 2016) connected with the particle diameter (Motos et al., 2019a) and with the formation of inorganic and organic coatings (Dalirian et al., 2018; Motos et al., 2019b)."** …

**P3L78 Suggestion to change order of words to "unprecedented vertically resolved airborne measurements"**

Changed following the reviewer suggestion.

**P16L484 NrBC-res is not in the figure. Did you mean NDro?**

Former Section 4 and former Figure 9 were removed.

**FIGURES**

**Figure 2b: The contribution to total mass of the largest size bins is not commented on in the text. Is this rBC or something else? How much of the mass do these particles represent?**

Regarding the larger bins of the size distribution.

- The last bin of the size distribution includes all particles showing an incandescence signal larger than saturation point of the detector. Since the diameter of particles associated with a saturated signal can not be quantified, it was set to be equal to the largest quantifiable diameter, in this case 575 nm.
- More details are given about the contribution of larger rBC particles to the total mass shown in Figure 2b and in Section 3.1:

  … **"These larger particles (mass geometric mean diameter above 400 nm) accounted for less than 5% of the total number concentration along the full altitude range. Nonetheless, they represented 37% of the total rBC mass observed in the lowest 500 m asl, and 17% in atmospheric layers aloft."** …

- Considering the size distributions shown in former Figure 4, the rBC particles contained in the last bin of the size distribution (501-575 nm) represented between 28-35% of the total rBC mass below-cloud and in-cloud, and less than 5% of the total rBC mass above-cloud.

  First, a note was added in the technical Section 2.2.3 to better explain the concept of signal saturation:

  … **"The rBC particles associated with a saturated incandesce signal were included in the largest bin of the size distribution (468 nm < $D_{rBC}$ ≤ 575 nm) and attributed with the maximum quantifiable mass-equivalent diameter (575 nm) and mass (178 fg)."** …

- A note was added in Section 3.3 to quantify the contribution of saturated signals to the total mass for the selected events above-cloud and below-cloud:

  … **"Compared to above-cloud, and increase of rBC particles larger than saturation-diameter (575 nm) was observed below-cloud. These larger particles represented less than 5% of $M_{rBC}$ above-cloud, and 32% of $M_{rBC}$ below-cloud."** …

- A note was added in Section 3.5.2 to quantify the contribution of saturated signals to the total mass for the selected events inside-cloud:

  … **"(rBC cores larger than 575 nm of mass-equivalent diameter, representing 28% of the $M_{rBC\text{-}res}$)"** …

Regarding the nature of the large particles. Since both reviewers raised this point, we address it in full details as it follows.

- COLOUR-RATIO: Different black carbon or soot types are characterized by a different boiling temperature, which can be derived from the ratio of the intensity of the incandescence signal detected by the broad-band detector (wavelength detection range of 300-800 nm) over the narrow-band detector (wavelength detection range of 630-800 nm). This ratio is commonly called "colour-ratio". Schwarz et al. (2006) showed how different soot types are characterized by different boiling-point temperatures. Dahlkötter (2014) showed an increase of the colour ratio from background conditions (values around 1.6-1.0) to biomass burning conditions (values around 2.0-1.5). So, the SP2 can be used to discriminate between different types of rBC.
- MINERAL-DUST INTERFERENCE: As shown by Schwarz et al. (2006), the SP2 is also sensitive to metal containing particles. This feature is nowadays exploited to derive the concentration of iron oxides with a modified SP2, based on non-overlapping detection range of two incandescence detectors (blue wavelength detection range of 300-550 nm; red wavelength detection range of 580-710 nm; Yoshida et al., 2016). According to the

570    latter, hematite and magnetite show a distinct colour ratio (~1.5) compared to rBC standard material (fullerene; 2.4). Considering the different wavelength detection range, the values presented by Yoshida et al. (2016) cannot be directly compared with ACLOUD or Dahlkötter (2014) measurements.

- COLOUR-RATIO DURING ACLOUD: The colour-ratio was calculated for the ACLOUD campaign: above-cloud median=1.49 (IQR=1.38-1.62), inside-cloud median=1.45 (IQR=1.31-1.66) and below-cloud median = 1.44 (IQR=1.30-1.66) cloud. The colour-ratio analysis was also performed as function of mas equivalent diameter (Figure B). Similar to Dahlkötter (2014), we observed a decrease of the colour-ratio with increasing rBC mass equivalent diameter. However, we did not observe a clear difference between the above-

580    cloud, inside-cloud and below-cloud measurements, which allowed for some clear screening of incandescing but yet non-rBC particles such as iron oxides as in Yoshida et al. (2016, 2020).

- SCREENING OF INVALID PARTICLES: Following the analysis performed in Yoshida et al. (2016), we tried to identify and remove those large particles (DrBC>300nm) showing prolonged incandescence signals, which might be related to mineral dust (Figure C). A signal was considered "valid" only if its duration was shorter than 30 us and the rise-time was shorter than decay time. Overall, a minor fraction of the total particles larger than 300 nm were considered to be invalid. The invalid particles did not show a peculiar colour-ratio which could be related to mineral dust and had a negligible effect on the size distributions

590    presented in the manuscript. Nonetheless, all the invalid particles were removed.

- TEXT CHANGES: A short statement was added to Section 2.2.3 and reads:

    … **"Previous studies showed that the SP2 is sensitive to metal containing particles such as hematite and magnetite, which might lead to an overestimation of rBC particle concentration (Schwarz et al., 2006; Yoshida et al., 2016). These particles are characterized by lower boiling point and colour-ratio (the ratio of thermal emission in the blue and red spectrum) but also by slow heating-rate in the laser beam of the SP2. During ACLOUD, particles associated with slow rise-time of the incandesce signal were removed. The colour-ratio analysis did not show any clear evidence of the presence of non-rBC yet incandescing particles."** …

600

[Figure]

*Figure E Diameter dependency of the color-ratio calculated for all incandesce signal observed above-cloud, inside-cloud and below-cloud with the SP2.*

[Figure]

*Figure F Incandescence signal of "valid" and "invalid" particle.*

**State in the figure caption that the in-cloud measurements were excluded.**
The caption was modified and now reads:
**… "rBC particles sampled behind the aerosol inlet outside clouds only and measured with the SP2"**
**…**

**Figure 3: Add to the text that this figure comprises "warm period" data only.**
The caption was modified and now reads:
**… "Atmospheric and cloud characterisation of four flights occurred on 2, 4, 5 and 8 June 2017 (warm-period) north-west of Svalbard." …**

**Figure 4. rBC size range does not match Table 1.**
We apologize for the mistake. The diameter quantification range of our SP2 during ACLOUD was 73-575 nm. The error was corrected in Table 1, in the caption of Figure 2,4,5,7,8, and everywhere else in the text. As a note, considering the scarcity of rBC particles inside and outside clouds, the size distribution presented in Figure 4 and 7 is built on a reduced number of bins (15), resulting in wider size-bins with minimum and maximum mid-bin diameter of 78-537nm.

**Figure 5. N_rBC-res is likely to be much higher since the SP2 does not measure particles below ~75? nm. Showing the mean/median number size distribution would help understand the degree of underestimation.**

- The reviewer can find the number size distribution of rBC particles in Figure G. From the figure is easy to understand that the number concentration of rBC particles presented in our work is clearly underestimated under every condition encountered during ACLOUD. In order to keep the manuscript short, Figure G was not included in the manuscript. Please note the change in the nomenclature for droplet number concentration from "$N_{Dro}$" to "$N_{Dro10}$" A clarification about $N_{rBC\text{-}res}/N_{Dro10}$ was added in Section 3.4.1:
  **… "Considering that the number size distribution culminated at the low quantification limit of the SP2; $N_{rBC\text{-}res}$ and, as a consequence, $N_{rBC\text{-}res}/N_{Dro10}$ were most certainly underestimated." …**

- The number concentration of rBC particles below the quantification limit of the SP2 could be estimated by fitting a lognormal function to the measured number size distribution. Due to the absence of a peak in the number size distribution, the estimated number concentration below the SP2's size-cut mostly depended on manual and arbitrary tweaking of the lognormal function constants. For this reason, a lognormal fit was not applied to extrapolate the number concentration below 73 nm. This technical aspect was addressed in the methodology Section 2.2.3:

[revised manuscript text omitted]

---

## Author Response (AR2)

**Acp-2023-30: "Airborne investigation of black carbon interaction with low-level, persistent, mixed-phase clouds in the Arctic summer"**

We would like to thank the referee and the editor for their comments. While the reviewer's and editor's comments are given in black bold, our answers are given below in grey letters. Additionally, we added the **changes made in the revised manuscript in grey bold letters**.

**Public justification (visible to the public if the article is accepted and published): I would like to thank the authors for incorporating the suggestions made by both reviewers. The revised version looks pretty good and it is almost ready for publication; however, there are some minor comments that need to be properly addressed before I can accept the manuscript.**

As following, we addressed the comments of the reviwer#2 and the editor. The major changes include:1) a full revision of grammar and syntax; 2) replacement of rBC with BC terminology, 3) implementation of colouring for colour-blindness in figures.

**Answers of the authors to Reviwer#2:**

**1, It is recommended to have the grammar of the entire text polished by a professional translation agency, for example:**
**Line 15: "a cloud nucleus"**
**Line 21: ", below,"**
**Line 21: "the increase in size"**
**Line 23: "a BC"**

The grammar was verified by a native English speaker.

**2, Line 101: It is mentioned here that there were 22 aircraft missions, but later in the text, it is clarified that only 17 missions' data were used. To avoid confusion, it is not necessary to mention 22 missions.**

Modified accordingly.

**3, Line 130: In line 121, it is stated that LWC is calculated based on the size distribution, but here it is mentioned that LWC is measured by SID-3. This may lead to ambiguity.**

Modified as:

... **"The mass fraction of ice water (IWF) was calculated from the IWC estimated by the CIP, and the LWC estimated by the SID-3." ...**

**4, Line 140: The use of "rBC" to represent BC measured by SP2 has a specific meaning according to Petzold et al. However, many subsequent studies on SP2 did not use this notation. This study did not extensively discuss the distinction between BC and rBC, nor did it discuss any differences in their interaction with clouds. Therefore, it is not recommended to use "rBC" instead use "BC" directly.**

The term "rBC" has was replaced with "BC" in the text, figures and tables. The following sentence was added in Section 2.2.3:

... **"The term refractory black carbon (BC) is used to identify the insoluble carbonaceous matter that vaporizes at temperatures around 4000 K, and that is measured with a laser-induced incandescence**

technique, including the SP2 (Petzold et al., 2013). To facilitate the reading, the term BC is used instead of rBC to identify all measurements performed with the SP2 and presented hereafter." …

**5, Line 153: There is a grammar issue with this sentence. It should be rewritten.**
The sentence now reads:
… "To estimate the rBC mass concentration outside the SP2's detection range, the rBC mass size distribution measure by the SP2 may be fitted with a lognormal fit (e.g. Laborde et al., 2013; Zanatta et al., 2018)." …

**6, Line 336: "...the results discussed as following are extremely uncertain..."**

**Line 433: "...unable to confirm nor to exclude the..."**
**The paper contains some unnecessary phrases and uses the term "might" extensively. It is important to be cautious with vague conclusions in academic papers.**
The text was revised to limit the use of vague sentences.

**Answers of the authors to the editor**

**1. There is a need to be consistent in the use of BC and rBC along the text.**
We revised the use of rBC and BC along the full manuscript, as also suggested by Reviewer#2.

**L17: Change "ACLOUD" with "Arctic CLoud Observations Using airborne measurements during polar Day (ACLOUD)"**
Modified accordingly.

L42-43: What do the authors mean with "its atmospheric layer"
We mean the atmospheric layer where BC is suspended. The statement was modified as:
… "causing a net warming of the local atmospheric layer (Flanner, 2013)" …

**L44, L83, L85, L106, L238, L436, L457, L459: "black carbon" should be "BC"**
Modified accordingly.

**L63: Add a reference after "schemes"**
Added Holopainen et al. (2020).

**L68: "lac" should be "lack"**
Modified accordingly.

**L71-73: The following text is unclear "If nucleation scavenging of aerosol particles from the below-cloud layer might represent the dominant activation mechanism in the Alaskan Arctic (Earle et al., 2011; McFarquhar et al., 2011), Igel et al. (2017) showed"**
The former long sentence was distilled in two parts:
… ". Nucleation scavenging of aerosol particles from the below-cloud layer represents the dominant activation mechanism in the Alaskan Arctic (Earle et al., 2011; McFarquhar et al., 2011). On the other hand, Igel et al. (2017) showed" …

**L81: "riming and Wegener" should be "riming and the Wegener"**
Modified accordingly.

**L88: "1) presence" should be "1) the presence"**
Modified accordingly.

**L89: "mechanisms; 3) impact" should be "mechanisms, and 3) the impact"**
Modified accordingly.

**L93: fix "(AC)3"**
Fixed as (AC)$^3$

**L103: "26 June" should be "26 June 2017"**
Modified accordingly.

**L104: "08 June" should be "08 June 2017"**
Modified accordingly.

**L107: "25 June" should be "25 June 2017"**
Modified accordingly.

**L110: "humidity and temperature" should be "humidity (RH) and temperature (T)"**
Modified accordingly.

**L121: "concentration (NDro) of liquid droplets" should be "concentration of liquid droplets (NDro)"**
Modified accordingly.

**L136: "27-29 June" should be "27-29 June 2017"**
Modified accordingly.

**L136: "13-17 June" should be "13-17 June 2017"**
Modified accordingly.

**L146: "differential mobility analyser" should be "scanning mobility particle sizer"**
Modified as
… "differential mobility analyser (DMA;" …

**L206: "SP2" should be "The SP2"**
Modified

**L237: Should "BC" be "rBC"?**
Modified accordingly.

**L247-248: Replace "-5.8 - -3.9°C" with "-5.8 to -3.9°C)"**
Modified accordingly.

**L256: "of Arctic boundary layer impacted cloud presence" is unclear**
Modified as:
… "representative of Arctic boundary layer influenced by cloud presence." …

**L259: Replace "where no" with "where neither"**

Modified accordingly.

**L268: Should "showed" be "shown"?**

Modified accordingly.

**L281-284: This part is unclear.**

The sentence was modified as:

… **"The size distribution of free-tropospheric rBC observed during ACLOUD is not uncommon in the Arctic spring and summer (Raatikainen et al., 2015; Taketani et al., 2016; Kodros et al., 2018; Zanatta et al., 2018; Schulz et al., 2019; Ohata et al., 2021). However, none of these previous Arctic studies ever reported rBC size distributions similar to below-cloud conditions."** …

**L285: I am not sure "All told" is appropriate.**

Modified in "overall"

**L299: I think "fund" should be "found"**

Modified accordingly.

**L315: I think "enriching" can be change by "increasing" or "enhancing"**

Modified accordingly.

**L336: Replace "as following" with "below".**

Modified accordingly.

**L347: "observation" should be "observations"**

Modified accordingly.

**L374-376: This part in unclear.**

The sentence was modified as:

… **"First, these results confirmed that larger and more hygroscopic rBC particles are usually enriched in cloud residuals (Motos et al., 2019). Second, the values above unity shown in** Error! Reference source not found.**, indicating an absolute enrichment of larger rBC-residuals compared to above-cloud and below-cloud, suggested the formation of these larger rBC as the result of in-cloud processing."** …

**L393: "The liquid water content" should be "LWC"**

Modified accordingly.

**L394: "The ice water content" should be "IWC"**

Modified accordingly.

**L427-431: This part is unclear.**

The sentence was modified as:

… **"Due to the low transmission efficiency of large drizzle drops in the CVI inlet, we were unable to verify the correlation between the diameter of rBC residuals and the concentration of drizzling drops. However, below-cloud release via evaporation (Igel et al., 2017) of rBC-agglomerates formerly contained in drizzling drops, and its reactivation at cloud-bottom (Solomon et al., 2015) might**

**contribute to the presence of larger rBC-residuals at cloud-bottom (**Error! Reference source not found.**) and explain the similarity between in-cloud and below-cloud size distribution (**Error! Reference source not found.**b,c).”** …

**Table 1: "this study. Including" should be "this study, including"**
Modified accordingly.

**Figure 1. I suggest to change BC to rBC in the entire figure.**
Modified accordingly.

**Figure 3: "rBC in cloud sampled behind the CVI inlet, otherwise behind the total inlet." This has to be grammatically improved.**
Modified accordingly.

**Figure 4: A), B), and C) labels are missing on the panels.**
Modified accordingly.

[revised manuscript text omitted]